# Human anogenital monocyte-derived dendritic cells and langerin+cDC2 are major HIV target cells

Jake W. Rhodes [1,2,15], Rachel A. Botting [1,2,3,15], Kirstie M. Bertram[1,2], Erica E. Vine [1,2], Hafsa Rana[1,2], Heeva Baharlou[1,2], Peter Vegh [3], Thomas R. O'Neil[1,2], Anneliese S. Ashhurst[4], James Fletcher[3], Grant P. Parnell[1,2], J. Dinny Graham[1,2], Najla Nasr[1,4], Jake J. K. Lim[5], Laith Barnouti[6], Peter Haertsch[7], Martijn P. Gosselink [1,8], Angelina Di Re [1,8], Faizur Reza[1,8], Grahame Ctercteko[1,8], Gregory J. Jenkins[9], Andrew J. Brooks[10], Ellis Patrick [1,11], Scott N. Byrne[1,4], Eric Hunter [12], Muzlifah A. Haniffa [3,13,14], Anthony L. Cunningham [1,2,16] & Andrew N. Harman [1,4,16 ✉]

Tissue mononuclear phagocytes (MNP) are specialised in pathogen detection and antigen presentation. As such they deliver HIV to its primary target cells; CD4 T cells. Most MNP HIV transmission studies have focused on epithelial MNPs. However, as mucosal trauma and inflammation are now known to be strongly associated with HIV transmission, here we examine the role of sub-epithelial MNPs which are present in a diverse array of subsets. We show that HIV can penetrate the epithelial surface to interact with sub-epithelial resident MNPs in anogenital explants and define the full array of subsets that are present in the human anogenital and colorectal tissues that HIV may encounter during sexual transmission. In doing so we identify two subsets that preferentially take up HIV, become infected and transmit the virus to CD4 T cells; CD14+CD1c+ monocyte-derived dendritic cells and langerin-expressing conventional dendritic cells 2 (cDC2).

[1] Centre for Virus Research, The Westmead Institute for Medical Research, Westmead, NSW, Australia. [2] Westmead Clinical School, Faculty of Medicine and Health Sydney, The University of Sydney, Westmead, NSW, Australia. [3] Biosciences Institute, The University of Newcastle, Newcastle upon Tyne, UK. [4] School of Medical Sciences, Faculty of Medicine and Health Sydney, The University of Sydney, Westmead, NSW, Australia. [5] Dr Jake Lim PLC, Parramatta, NSW, Australia. [6] Australia Plastic Surgery, Sydney, NSW, Australia. [7] Burns Unit, Concord Repatriation General Hospital, Sydney, Australia. [8] Department of Colorectal Surgery, Westmead Hospital, Westmead, NSW, Australia. [9] Department of Obstetrics and Gynaecology, Westmead Hospital, Westmead, NSW, Australia. [10] Department of Urology, Westmead Hospital, Westmead, NSW, Australia. [11] School of Maths and Statistics, Faculty of Science, The University of Sydney, Camperdown, NSW, Australia. [12] Emory Vaccine Centre, Atlanta, GA, USA. [13] Wellcome Sanger Institute, Hinxton, UK. [14] Department of Dermatology and NIHR Newcastle Biomedical Research Centre, Newcastle Hospitals NHS Foundation Trust, Newcastle upon Tyne, UK. [15] These authors contributed equally: Jake W. Rhodes, Rachel A. Botting. [16] These authors jointly supervised this work: Anthony L. Cunningham, Andrew N. Harman. ✉email: andrew.harman@sydney.edu.au

There is still no cure or vaccine for HIV/AIDS and 37 million people remain infected. Antiretroviral therapy (ART) is efficient at controlling infection but is a lifelong treatment which is costly to manage and associated with toxicities. Only 57% of HIV[+] individuals receive ART and there are 1.8 million new infections each year. ART can be given to healthy 'at risk' individuals as pre-exposure prophylaxis (PrEP) which has shown to be effective in reducing transmission. However, this is not a universal solution because of poor access to PrEP in low income countries and variable uptake in Western countries. Furthermore, the effects of long-term administration of PrEP to healthy individuals are unknown and can be associated with decreased condom use[1], increased sexually transmitted infections and concomitant genital tract inflammation[2], enhancing HIV transmission[3–6], especially in sub-Saharan Africa. Furthermore, PrEP regimens have recently been shown to be ineffective in the context of an inflamed mucosa[7–9]. Therefore, an effective vaccine and cure are still needed.

HIV is now transmitted sexually in almost all cases. In order to develop a vaccine (or more effective PrEP regimens) the precise definition of the initial HIV target cells in the anogenital mucosa is necessary, especially mononuclear phagocytes (MNP). These consist of Langerhans cells (LC), dendritic cells (DC) and macrophages which express the HIV entry receptors CD4 and CCR5 allowing them to be directly infected. Importantly, they also express a large repertoire of lectin receptors including C-Type Lectins (CLR) and Sialic acid-binding immunoglobulin-type lectins (Siglec), many of which can bind HIV and mediate endocytic uptake of the virus[10–15]. As professional antigen presenting cells, LCs and DCs play a critical role in HIV transmission by transferring the virus to CD4 T cells[13,16,17] where it replicates resulting in CD4 T cell death and depletion and consequent immunosuppression. Epidermal LC have been shown to take up HIV and transfer it to T cells in vagina, cervix and foreskin[18–20] and we and others have recently shown that epidermal DC populations also participate in this process[21,22]. We also previously showed that HIV transfer to CD4 T cells occurs in two successive phases from epidermal LCs[13], epidermal CD11c[+] DCs[21] and in vitro derived monocyte-derived (MDDC)[17,23]. The first phase of transfer occurs within 2 h and is dependent on lectin mediated uptake of the virus and declines rapidly with time. The second phase occurs from 72 h onwards and increases with time as newly formed virions bud off from the surface of cells that have become productively infected via CD4/CCR5 mediated entry into viral synapses[17].

Mucosal trauma which breaches anogenital and colorectal epithelium and associated inflammation are likely to enhance HIV acquisition as it also allows the virus direct access to deeper target cells in the underlying lamina propria. Despite this, the role of lamina propria MNP subsets in HIV transmission has been largely understudied. These MNPs are present in several distinct subsets which include cDC1[24,25] and cDC2[26]. In addition, there are several subsets of CD14[+] cells[27–31] which include autofluorescent tissue resident macrophages[26] and non-autofluorescent CD14[+] cells which have been conventionally referred to as DCs[26] and have an established role in transmitting HIV to CD4 T cells in both intestinal[32–34] and cervical[15] tissue. However, in healthy tissue CD14[+]CD1c[−] cells have recently been redefined as monocyte-derived macrophages (MDM)[35]. This makes their role in HIV transmission puzzling as macrophages are very weak antigen presenting cells for naïve T cells and do not migrate to lymph nodes and thus are less likely to deliver the virus to CD4 T cells than DCs. In inflamed tissue CD14[+]CD1c[+] cells have been described which transcriptionally align with DCs[36,37] and have been defined as in vivo MDDCs[31]. Recently similar cells have also been described in healthy lung tissue[38].

In this study we have thoroughly defined the MNP subsets that are present in the sub-epithelium (lamina propria and dermis) of all human anogenital and colorectal tissues and found two cell subsets that are key players in HIV uptake and transmission; CD14[+]CD1c[+] ex vivo MDDCs and langerin[+] cDC2. Compared to other MNPs both subsets took up HIV more efficiently and became more infected. We found that ex vivo CD14[+]CD1c[+] MDDCs were most efficient at transferring HIV to CD4 T cells at late time points which correlated with their high CCR5 expression and that langerin[+] cDC2 transferred the virus most efficiently at early time points.

## Results

**Defining human mucosal mononuclear phagocyte subsets by flow cytometry.** We have previously optimised protocols for the efficient isolation and definition of MNPs from human skin with minimal surface receptor cleavage and in an immature state to most closely resemble their functional state when they encounter pathogens[39]. We firstly modified our skin flow cytometry gating strategy for use in mucosal tissues (Fig. 1A). All known tissue MNPs were identified which were; epidermal CD11c[+] DCs[21] and LCs and dermal cDC1, cDC2, CD14[+] autofluorescent macrophages and dermal non-autofluorescent CD14[+] cells. We also designed the panel to differentiate between non-autofluorescent ex vivo CD14[+]CD1c[−] MDMs[40] and ex vivo CD14[+]CD1c[+] MDDCs[30,31,37]. We used this gating strategy to define the relative proportions of each dermal MNP subset in the full range of human anogenital and colorectal tissues that are the actual sites where HIV transmission occurs, as well as abdominal skin for comparison. This included tissues comprised of skin (labia, outer foreskin, glans penis and anal verge), Type II mucosa (vagina, fossa navicularis, ectocervix and anal canal) and type I mucosa (endocervix, penile urethra, rectum and colon) (Fig. 1B). In order to confirm these trends in situ we also used fluorescence microscopy to show similar trends in cell population density using inner foreskin (Fig. 2A). The relative proportions of cDC1 was relatively consistent across all tissues with the exception of outer foreskin which had slightly higher proportions. In abdominal skin we found that dermal cDC2 were the overwhelmingly predominant cell population in contrast anogenital and colorectal tissues where these cells were present in much smaller proportions. Notably, however, a greater proportion of the total cDC2 population expressed langerin in these tissues compared to abdomen (Fig. 2B). Conversely, CD14-expressing cells were present in higher proportions in anogenital and colorectal tissues compared to abdomen and the relative proportions were significantly higher in mucosal tissues compared to anogenital skin (Fig. 2C). Despite the fact that in vivo MDDCs have predominantly been described as an inflammatory cell we found that they were substantially present across all our uninflamed tissues.

**Transcriptional profiling.** As CD14[+] cells have been shown to play an important role in HIV transmission we firstly focused our study on these cells. As we found both ex vivo CD14[+]CD1c[+] MDDC and ex vivo CD14[−]CD1c[−] MDM in all our uninflamed tissue samples we carried out RNAseq analysis to compare the transcription profile of these cells to other tissue MNPs; LCs, dermal langerin[−] cDC2 and their langerin-expressing counterparts (Fig. 3A). As expected, LCs were the most distinct population and, in agreement with the literature, cDC2 clustered together regardless of langerin expression[41] and CD14[+]CD1c[−] cells were transcriptionally very similar to autofluorescent macrophages[35] whereas CD14[+]CD1c[+] cells aligned more closely with DCs[37,38]. Comparison of ex vivo MDM cells with ex vivo MDDC cells showed 501 differentially expressed genes. Ex vivo

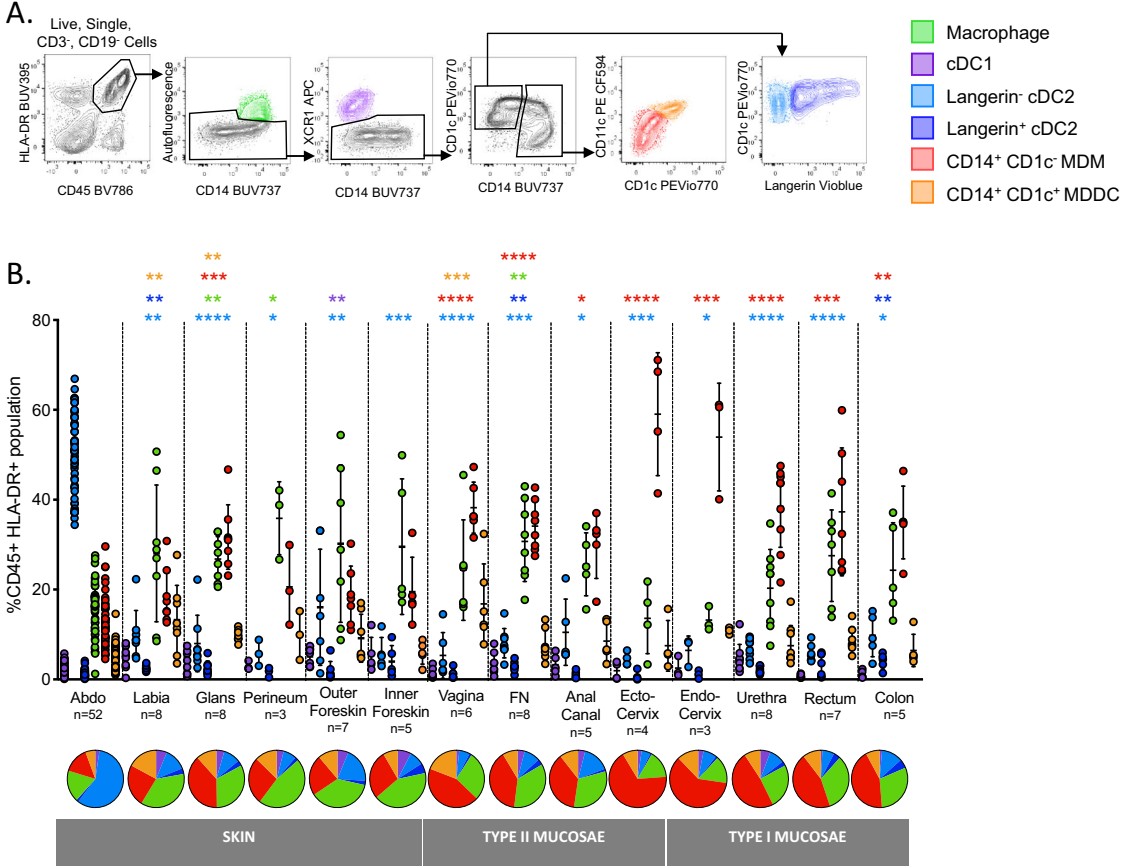

**Fig. 1 Definition of human dermal/lamina propria mononuclear phagocytes by flow cytometry. A** Following collagenase digestion six distinct CD3−CD19− CD45+HLA-DR+ mononuclear phagocyte subsets from human abdominal skin were defined, (i) tissue resident macrophages (green) were defined as autofluorescent+CD14+, (ii) type 1 conventional dendritic cells (cDC1; purple) were defined as autofluorescent−, XCR1+, CD14−, type 2 conventional dendritic cells (cDC2; blue) were defined as autofluorescent−, XCR1−, CD14−, CD1c+, and could be split into two populations, (iii) a langerin− population (light blue) and (iv) a langerin+ population (dark blue), (v) CD14+CD1c− cells (red) were defined as autofluorescent−, XCR1−, CD14+, CD1c−, (vi) CD14+ CD1c+ cells (orange) were defined as autofluorescent−, XCR1−, CD14+, CD1c+. Representative plot of n = 52 abdominal donors is shown. **B** Relative proportions of each subset of mononuclear phagocyte as a percentage of CD45+ HLA-DR+ gate across the human anogenital/colorectal tracts were determined and mean ± standard deviation plotted. Statistics for subsets in each tissue were generated using the Kruskal–Wallis test: two-tailed Dunn's multiple comparisons, comparing against abdominal skin tissue. *p < 0.05; **p < 0.01; ***p < 0.001; ****p < 0.0001. (abdominal tissue (Abdo) = 52, labia = 8, glans penis = 8, perineum = 3, outer foreskin = 7, inner foreskin = 5, vagina = 6, fossa navicularis (FN) = 8, anal canal = 5, ectocervix = 4, endocervix = 3, penile urethra = 8, rectum = 7, colon = 5). Underneath, a pie chart for each tissue shows the mean proportion of each dermal/lamina propria subset across the human anogenital/colorectal tracts.

MDDCs expressed lower levels of the lentiviral restriction factor *APOBEC3G* which is a known inhibitor of HIV-1 infectivity[42]. We also measured the protein expression levels of SAMHD1 given its known role as a myeloid HIV restriction factor. It was expressed at slightly highly levels in ex vivo MDDCs than in ex vivo MDMs although this was not statistically significant. However, it was expressed at significantly higher levels in cDC1 (Fig. 3B).

**Determination of surface receptor expression on skin mononuclear phagocytes.** Previously we have carried out gene expression studies to examine CLR expression by ex vivo-derived skin DC subsets derived by collagenase digestion and compared them to model in vitro derived cells[43]. However, we did not include cDC1 or autofluorescent macrophages nor did we divide the non-autofluorescent CD14-expressing cells according to CD1c expression or cDC2 by langerin expression. Importantly, we also did not examine the surface expression of these proteins. We therefore used our flow cytometry gating strategy (Fig. 1A) to determine the surface expression levels of a wide range of surface

molecules including HIV entry receptors, costimulatory molecules and lectin receptors involved in pathogen recognition (CLRs and Siglecs) on each MNP subset obtained via enzymatic digestion (for immature cells) or spontaneous migration (for mature cells). We also included MDDC and MDM derived in vitro from blood CD14 monocytes which are commonly used as model cells (Fig. 3C). Importantly, we used appropriate collagenase enzyme blends which we have previously determined do not cleave each specific surface receptor[39]. As we had carried out RNAseq for cells derived by enzymatic digestion we were able to compare gene and surface expression profiles for immature cells isolated by this method. In almost all cases similar trends were observed (Fig. 3D). Discrepancies between mRNA and surface expression levels were detected in molecules upregulated due to MNP maturation (*CD80*, *CD83* and *CXCR4*). This is most likely due to the fact that the process of tissue isolation triggers maturations as shown previously[39]. Thus, at the time point we measured surface expression (immediately after isolation) changes in gene expression had not yet translated to surface expression.

All MNPs expressed the key HIV entry receptor CD4 but cDC1 and LCs expressed very low levels of the HIV entry co-receptor

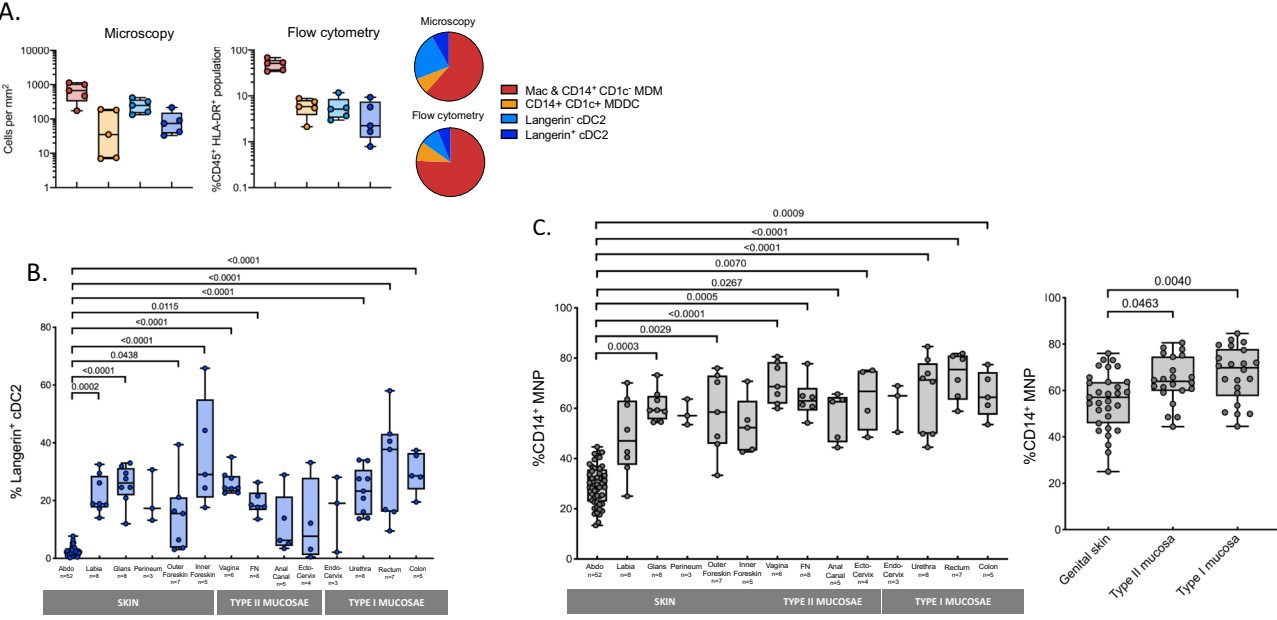

**Fig. 2 Proportion of cDC2 and CD14-expressing mononuclear phagocytes in human sub-epithelial tissue. A** Using fluorescent microscopy the proportion of sub-epithelial MNP subsets were quantified in inner foreskin ($n = 5$) to a depth of 60 μm from the basement membrane and compared to the proportion quantified using flow cytometry ($n = 5$). Plotted as box and whisker plots, box representing the upper and lower quartile, central line representing the median, the whiskers the minimum and maximum of each cell subset and each donor represented by an individual dot. Pie charts show the mean proportion of MNP subsets using both quantification methods. Combined CD14+ CD1c− monocyte-derived macrophages (MDM) and macrophages (Mac) (red), CD14+ CD1c+ monocyte-derived dendritic cells (MDDC) (orange), langerin- cDC2 (light blue) and langerin+ cDC2 (dark blue). **B** The proportion of langerin+ cDC2 across the human anogenital/colorectal tissue plotted as box and whisker plots, box representing the upper and lower quartile, central line representing the median, the whiskers the minimum and maximum of each tissue sample and each donor represented by an individual dot. Statistics were generated using the Kruskal–Wallis test: two-tailed Dunn's multiple comparisons with adjusted P values shown. (abdominal tissue (Abdo) = 52, labia = 8, glans penis = 8, perineum = 3, outer foreskin = 7, inner foreskin = 5, vagina = 6, fossa navicularis (FN) = 8, anal canal = 5, ectocervix = 4, endocervix = 3, penile urethra = 8, rectum = 7, colon = 5). **C** The proportion of CD14+ mononuclear phagocytes (macrophages, CD14+CD1c− cells, CD14+CD1c+ cells) across the human anogenital/colorectal tract. Left: looking at individual anogenital/colorectal tissue (abdominal tissue (Abdo) = 52, labia = 8, glans penis = 8, perineum = 3, outer foreskin = 7, inner foreskin = 5, vagina = 6, fossa navicularis (FN) = 8, anal canal = 5, ectocervix = 4, endocervix = 3, penile urethra = 8, rectum = 6, colon = 5), right: pooled data for genital skin (labia, glans penis, perineum, outer foreskin, inner foreskin; $n = 31$) type II mucosae (vagina, fossa navicularis, anal canal, ectocervix; $n = 22$) and type I mucosae (endocervix, penile urethra, rectum, colon $n = 22$) plotted as box and whisker plots, box representing the upper and lower quartile, central line representing the median, the whiskers the minimum and maximum of each tissue sample and each donor represented by an individual dot. Statistics as described above for **B**.

CCR5. These two subsets also differed from other MNPs subsets in that they expressed far fewer lectin receptors. LCs only expressed langerin, DCIR, DEC205, Siglec-3 and Siglec-9 as well as very low levels of CLEC7A, CLEC4D and CLEC4E. cDC1 expressed slightly more lectin receptors than LCs but generally at lower levels than other MNPs and uniquely expressed CLEC9A consistent with the literature. Other than langerin expression, we did not identify any differences in surface expression molecules between langerin-expressing and non-expressing cDC2.

We next focused our attention on key differences in the CLR profiles between DC like cells and macrophage like cells. Inflammatory MDDCs have been reported to express CD141 and CD1a[31,37] but we showed these were also not expressed on the surface our cells derived from uninflamed tissue (Fig. 3C). Interestingly, the HIV binding CLR DC-SIGN, previously thought to be a DC marker and shown to be involved in capture of HIV and transfer to T cells[11,31–33,44–47], was not expressed by any ex vivo DC subsets or ex vivo CD14+CD1c+ MDDCs (although gene expression was detected). It was expressed highly by ex vivo autofluorescent macrophages and in vitro derived MDDCs and at low levels by both ex vivo CD14+CD1c− and in vitro MDMs (Fig. 3C, D). Similarly Siglec-1, a lectin shown to be important in DC-mediated HIV uptake[15,48,49] was expressed most highly by autofluorescent macrophages and ex vivo MDMs but at lower levels on ex vivo MDDCs lower still on cDC1 and

cDC2 and not at all by in vitro derived MDDCs (Fig. 3C, D). Finally, CLEC5A was expressed very highly by cDC2 and ex vivo MDDCs but at lower levels on other CD14+ cells. Other than this, all CD14+ ex vivo derived cells showed very similar surface expression profiles. Taken together the expression profiles of DC-SIGN, Siglec-1 and CLEC5A add further weight to the hypothesis that CD14+CD1c+ cells are DC-like and CD14+CD1c− cells are macrophage like.

In addition to what is described above, model in vitro derived MDDCs and MDM differed from bona fide ex vivo derived cells in several important ways; similar to LCs, they did not express L-SIGN, CLEC7A, CLEC10A and Siglec-6 but, unlike LCs did not express CLEC4D or langerin. They also expressed much higher levels of CD4 and DEC205 than any other cell type.

Finally, we investigated differences in surface receptors between immature and mature cells. Using the spontaneous migration method we were only able to reliably isolate langerin− cDC2 and ex vivo CD14+CD1c+ MDDCs as described previously[39] (Fig. 3E). As expected, cells isolated by this method expressed much higher levels of CD80 and CD86 than cells isolated by enzymatic digestion confirming their mature state. Interestingly cDC2, but not ex vivo MDDCs expressed CD83. Cells isolated by spontaneous migration expressed much lower (or none) of the following molecules: CLEC4A, CLEC4K (langerin), CLEC4L (DC-SIGN), CLEC8A, CLEC10A, CLEC12A and Siglec-1, while

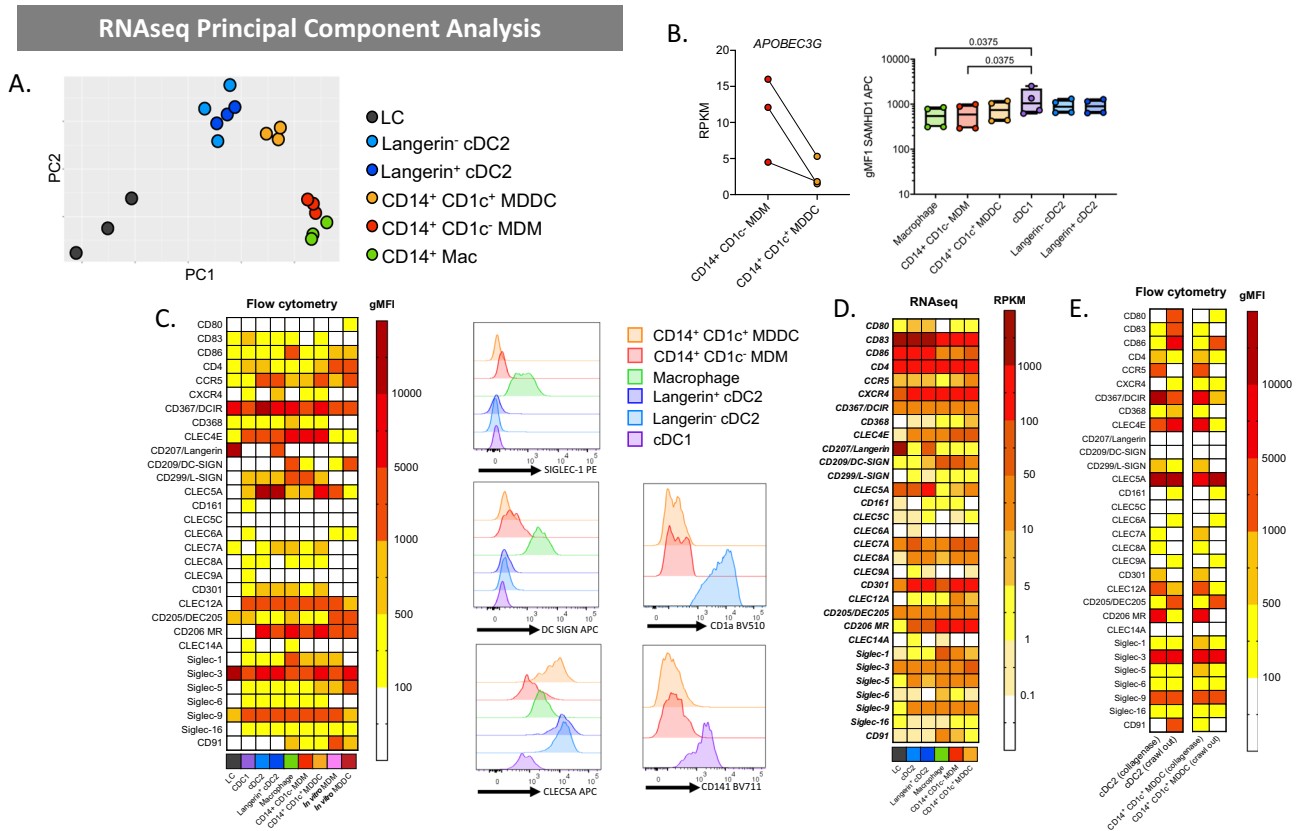

**Fig. 3 Genotypic and phenotypic profiling of epidermal and dermal human abdominal skin mononuclear phagocytes. A** Dermal macrophages, cDC2 (langerin− and langerin+), CD14+CD1c+ cells and CD14+CD1c− cells as well as epidermal Langerhans cells were liberated from $n = 3$ abdominal skin donors and FACS isolated for RNAsequencing (cDC1 were omitted from analysis due to low donor numbers). A PCA plot shows clustering of cell subsets. **B** Left: HIV restriction factor APOBEC3G gene expression by RNAseq for CD14+CD1c− and CD14+CD1c+ cells ($n = 3$). Reads per kilobase of transcript, per million mapped reads (RPKM) is plotted with each dot representing an individual donor and lines matching donors. Rights: intracellular SAMHD1 expression on MNP subsets from abdominal skin assessed by flow cytometry, plotted as box and whisker plots, box representing the upper and lower quartile, central line representing the median, the whiskers the minimum and maximum of each tissue sample and each donor represented by an individual dot. Statistics generated using Friedman test with two-tailed Dunn's multiple comparisons adjusted $P$ values shown ($n = 5$). **C** Epidermal LCs and dermal MNPs were liberated from tissue using enzymatic digestions and surface receptor expression evaluated by flow cytometry and compared to in vitro monocyte-derived macrophages (MDM) and monocyte-derived dendritic cells (MDDC). Left: the geometric mean fluorescent intensity (gMFI) minus the isotype for each subset was calculated and mean plotted in a heat map ($n = 2$–11). Right: representative histograms of one donor for cell surface expression of three c-type lectin receptors of interest, Siglec-1, DC-SIGN, CLEC5a for all subsets and CD1a and CD141 expression on CD14+CD1c− cells and CD14+CD1c+ compared to cDC2 and cDC1 respectively. **D** A heatmap was also generated from the RNAseq data to investigate the expression of the corresponding genes investigated by flow cytometry in **C**. **E** Flow cytometry analysis shows differences in cell surface expression on immature (enzymatically digested-derived cells) vs mature (spontaneous migration-derived cells) for left: cDC2 and right: CD14+CD1c+ cells. LC (black), cDC1 (purple), langerin− cDC2 (light blue), langerin+ cDC2 (dark blue), dermal macrophages (green), CD14+ CD1c− MDM (red), CD14+ CD1c+ MDDC (orange), in vitro MDM (pink) and in vitro MDDC (crimson).

CLEC4E was down regulated by ex vivo MDDCs but not cDC2. Conversely CLEC13B (DEC205) was upregulated. In terms of HIV entry receptors all cell subsets expressed CD4 regardless of their method of isolation. As we showed previously, CCR5 expression was not detected on immature LCs[21] or cDC1 or any mature cells. CXCR4 was expressed at very low levels or not at all by mature cells but was expressed at slightly higher levels by LCs and ex vivo MDDCs isolated by the spontaneous migration method.

**CD14+ dermal cell morphology and tissue residency.** As DCs and macrophages are morphologically different[26] we next compared the morphology of dermal ex vivo MDDCs, ex vivo MDMs and cDC2 using Giemsa staining (Fig. 4A). Consistent with the transcriptional profiling and surface expression profile, we found that ex vivo MDMs looked like macrophages with a 'fried egg' like appearance and containing many large intracellular vacuoles

whereas ex vivo MDDCs cells were morphologically more similar to DCs with none or few vacuoles present.

DC and macrophages also differ in tissue residency which has important implications for transmission of HIV to CD4 T cells; DCs migrate out of tissue to lymph nodes which are rich in CD4 T cells whereas macrophages remain tissue resident. Consistent with this migration differential, DCs can be isolated from tissues by the spontaneous migration method (albeit in an activated state)[39] whereas macrophage isolation requires enzymatic digestion. We therefore compared the ability of dermal ex vivo MDDCs and ex vivo MDMs to migrate out of tissue spontaneously (Fig. 4B). Consistent with their DC-like transcriptional and morphological phenotype ex vivo MDDCs migrated out of tissue whereas ex vivo MDMs did not, consistent with their macrophage like phenotype. To confirm that the ex vivo MDMs were still present within the tissue, we digested the tissue after spontaneous migration and found that they were still present. We

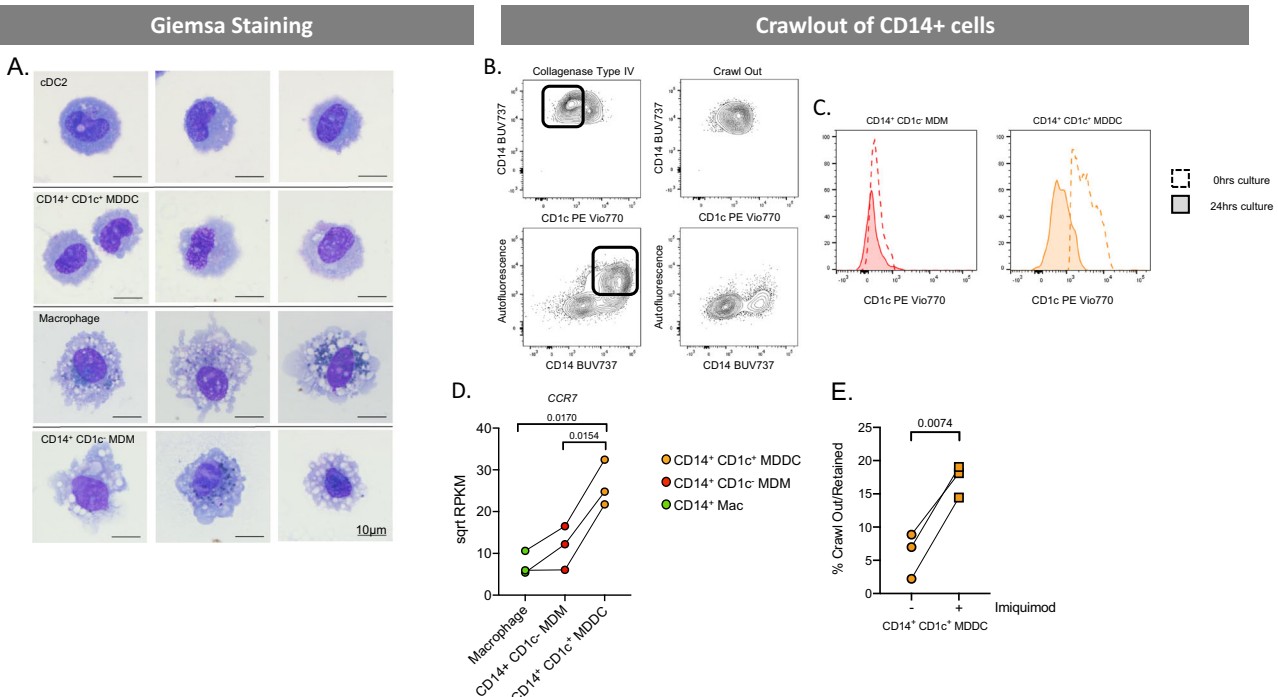

**Fig. 4 Morphological and migratory properties of CD14-expressing human tissue mononuclear phagocytes. A** Human dermal MNPs were isolated from abdominal skin by collagenase digestion, with each defined subset FACS isolated and Giemsa stained to investigate morphology. **B** The migratory capabilities of CD14-expressing cells (macrophages, CD14$^+$CD1c$^-$ cells, CD14$^+$CD1c$^-$ cells). From a single donor dermal sheets were cultured for 24 h and supernatant collected for cells which had undergone spontaneous migration (crawl out), tissue was then collagenase digested to liberate cells which did not undergo spontaneous migration. Contour plots show the phenotype of cells which crawled out compared to those which remained within the tissue, box highlighting cells which remained in tissue. **C** From a separate donor CD14$^+$CD1c$^-$ cells and CD14$^+$CD1c$^+$ cells were FACS isolated following collagenase digestion and cultured for 0 h (unfilled, dotted histogram) and 24 h (filled, continuous histogram) to determine the difference in CD1c expression on these cells before and after maturation. **D** Gene expression of chemokine receptor CCR7 determined by RNAseq for CD14-expressing MNPs from abdominal skin ($n = 3$). The square root of reads per kilobase of transcript, per million mapped reads (RPKM) is plotted, with each donor represented by an individual dot and lines connecting donors. Statistics were generated using RM one-way ANOVA with Holm–Sidak's multiple comparisons with adjusted $P$ values shown. **E** The percentage of CD14$^+$CD1c$^+$ cells which had undergone spontaneous migration compared to cells which remained within the tissue was calculated following overnight culture with or without the TLR7 agonist imiquimod. Statistics were calculated using a two-tailed paired $T$ test ($n = 3$). Macrophages (green), CD14$^+$ CD1c$^-$ MDM (red) and CD14$^+$ CD1c$^+$ (orange).

next sorted ex vivo MDM and ex vivo MDDCs after isolation by enzymatic digestion and confirmed that ex vivo MDMs did not upregulate CD1c with culture, whereas ex vivo MDDCs showed a slight downregulation of CD1c, suggesting the population of cells which migrated out of tissue were bona fide CD14$^+$CD1c$^+$ MDDCs (Fig. 4C). As CCR7 is a key chemokine receptor required for DC migration out of tissue and shown in lung to be expressed by ex vivo MDDC[38], we compared the CCR7 gene expression levels in both cells types and showed that CCR7 was expressed much more highly on ex vivo MDDCs (Fig. 4D). Finally, as it is postulated that HIV or other stimuli, such as bacterial components leaked through damaged epithelium due to local trauma, would lead to activation of DCs and migration of HIV-laden DCs, we stimulated abdominal skin tissue with the TLR agonist imiquimod and showed that this significantly increased the number of ex vivo MDDCs that migrated out of the tissue (Fig. 4E). Taken together this shows that ex vivo MDDCs migrate out of tissue whereas ex vivo MDMs do not.

**CD14$^+$ dermal cell MNPs take up HIV via Siglec-1.** We next compared the ability of ex vivo derived CD14-expressing cell subsets to take up HIV after 2 h of exposure using both the lab-adapted Bal strain and the transmitted/founder Z3678M strain[21,50]. All subsets efficiently took up the virus but ex vivo MDDCs took up significantly more (Fig. 5A). We were also able

to observe both ex vivo MDDCs and ex vivo MDMs interacting with HIV in situ in penile urethra using our RNAscope HIV detection assay[21] indicating that HIV can penetrate into the sub-epithelium and interact with these cells in situ (Fig. 5B). CD14-expressing cells have previously been shown to play a role in HIV transmission and we and others have shown that DC-SIGN is a key HIV binding receptor expressed on these cells and is necessary for uptake and transfer to CD4 T cells[11,32–34,44–46,51]. Furthermore we have shown blocking the ability of HIV to bind both DC-SIGN and mannose receptor on MDDCs derived in vitro from blood CD14$^+$ monocytes significantly blocks HIV uptake[11,12]. However, we observed here that another key HIV binding lectin receptor, Siglec-1[15,47–49], was expressed by all CD14$^+$ sub-epithelial cells and at much higher levels than CD14$^-$ MNP subsets (Fig. 3C). We therefore examined Siglec-1 expression on HIV infected and by stander cells derived from labia, inner foreskin and colon. In all CD14-expressing cell types we found that cells that had taken up HIV expressed significantly higher levels of Siglec-1 than cells that did not take up HIV, indicating that Siglec-1 expression correlates with HIV uptake (Fig. 5C). Furthermore, in all specimens Siglec-1 was expressed most highly in autofluorescent tissue resident macrophages followed by ex vivo MDM then ex vivo MDDCs. This finding was confirmed in colonic tissue (Fig. 5D). Using cells derived from labia, foreskin and colon we showed that a Siglec-1 antibody was able to partially block HIV uptake further implying a role of

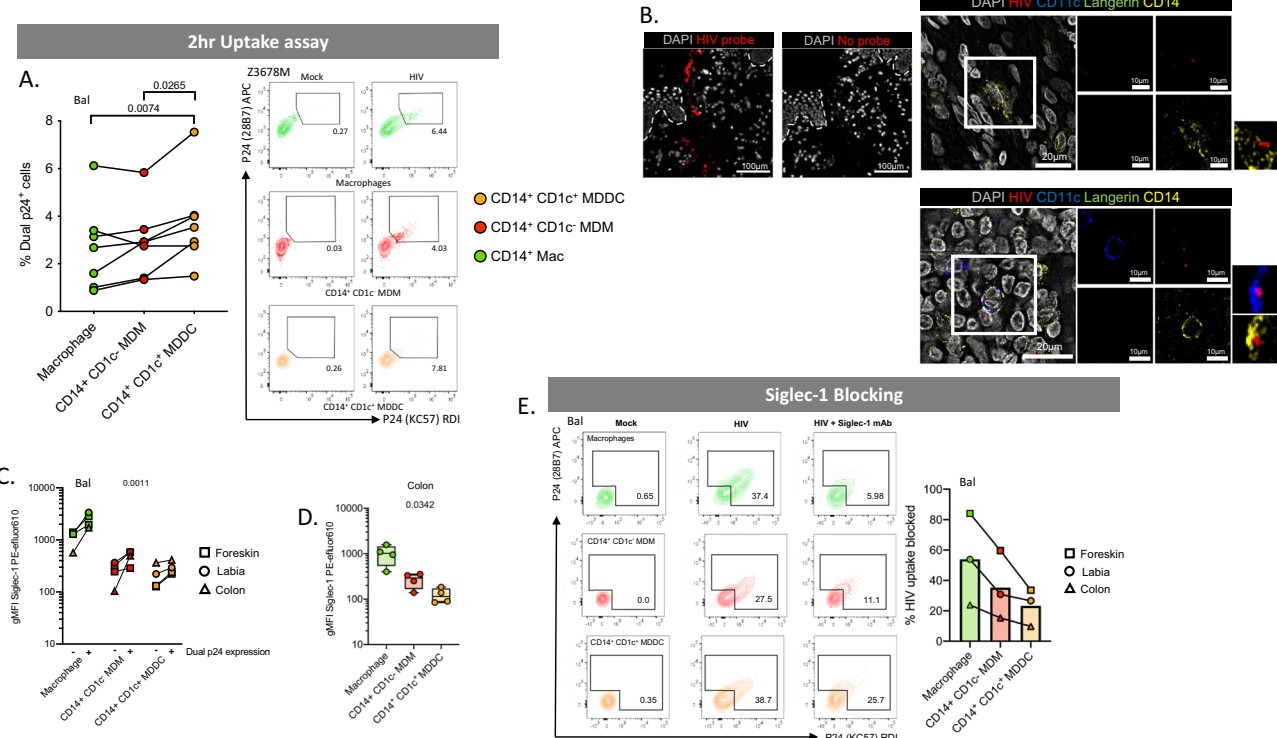

**Fig. 5 HIV uptake of dermal CD14-expressing mononuclear phagocytes. A** MNPs were liberated from human abdominal skin using and CD45+HLA-DR+ live cells FACS isolated. Mixed dermal populations were incubated for 2 h with HIV_Bal or HIV_Z3678M or mock treated, thoroughly washed and stained for surface markers and two antibody clones to HIV p24 (KC57 and 28B7) for flow cytometry analysis to determine percentage dual p24+ cells. Left: HIV_Bal results graphed with each donor represented by individual dots, donor matched by connected lines ($n = 7$). Statistics were generated using RM one-way ANOVA with Holm-Sidak's multiple comparisons with adjusted $P$ values shown. Right: representative contour plots for HIV_Z3678M shown, gating on dual p24+ cells for each subset. **B** Human penile urethra explants were treated with HIV for 2 h before being fixed and paraffin embedded. After sectioning, HIV RNA was visualised using RNAscope and immunofluorescent staining for langerin, CD11c, CD14 and DAPI. Left: representative images of one donor, with and without the addition of the HIV specific RNAscope probe. Right: macrophages and CD14+CD1c− cells could not be individually distinguished from each other but could be visualised as a group (top) while CD14+CD1c+ cells could also be visualised interacting with HIV (red) (bottom). Representative images from $n = 3$ donors. **C** Human MNPs from anogenital/colorectal tissue (2x foreskin: squares, 1x labia: circle, 1x colon: triangle), were treated as in **A**. The expression of Siglec-1 was assessed by geometric mean fluorescence intensity (gMFI) minus matching FMO control, on dual p24+ cells vs p24− cells. Each dot represents an individual donor with lines matching the same donor for each subset. Statistics measured the difference in p24+ cells versus p24− cells using a three-way ANOVA. **D** Siglec-1 expression on CD14-expressing subsets was determined in $n = 4$ colon donors. plotted as box and whisker plots, box representing the upper and lower quartile, central line representing the median, the whiskers the minimum and maximum of each cell subset and each donor represented by an individual dot. Statistics generated using RM one-way ANOVA. **E** Mixed dermal MNPs were treated as above however before incubation with HIV, cells were pre-treated with Siglec-1 mAb for 30 min. Left: representative plot of HIV uptake assessed by dual p24+ cells in Mock, HIV treated and HIV + Siglec-1 mAb samples from foreskin donor. Right: percentage of HIV uptake that was blocked using Siglec-1 mAb across three donors of anogenital/colorectal tissue (foreskin: square, labia: circle, colon: triangle), with each point representing individual donors matched with joining lines, and column representing mean. Macrophages (green), CD14+ CD1c− MDM (red) and CD14+ CD1c+ (orange).

Siglec-1 in HIV uptake by CD14-expressing cells (Fig. 5E). We also measured the ability of all subsets derived from abdominal skin to transfer HIV to the JLTR cell line at 2 h (Fig. 6A). We then confirmed that the combined CD14-expressing cells derived from abdominal tissue and ex vivo MDDCs derived from colonic tissue could transfer HIV to activated primary CD4 T cells derived from PBMCs (Fig. 6B). We found no statistically significant difference between the three subsets' ability to transfer the virus, although autofluorescent tissue resident macrophages (which expressed the highest levels of Siglec-1) were able to transfer the virus much more efficiently in 2 of the 4 donors. Therefore, Siglec-1 expression does not explain why ex vivo MDDCs cells take up HIV more efficiently than other CD14-expressing cells.

**Ex vivo MDDCs are most susceptible to HIV infection.** We next investigated the ability of dermal CD14-expressing cells to become productively infected with HIV and to transfer the virus to CD4 T cells at later time points. As the expression of the HIV

entry receptors CD4 and CCR5 are essential for productive HIV infection, we firstly measured the surface expression of these molecules on ex vivo cells from abdominal skin and found that ex vivo MDDCs expressed higher levels of CCR5 than other CD14-expressing populations (Fig. 7A). We then confirmed this observation in cells derived from colon tissue which is more relevant to HIV transmission (Fig. 7B). Corresponding with their higher CCR5 surface expression, we also found that supernatants derived from infected abdominal ex vivo MDDCs cultures were able to infect greater numbers of CD4 TZMBL cells than other CD14-expressing subsets (Fig. 7C). To confirm that cells were productively infected at these late time points (as opposed to virus being held on the surface) we infected CD4 T cells with supernatants derived from combined CD14+ cells that had been treated with HIV in the presence of AZT and showed that a reduction in the number of CD4 T cells that became infected (Fig. 7D). Despite using our optimised ex vivo derived skin MNP culture methods[39], we were unable to measure direct infection of

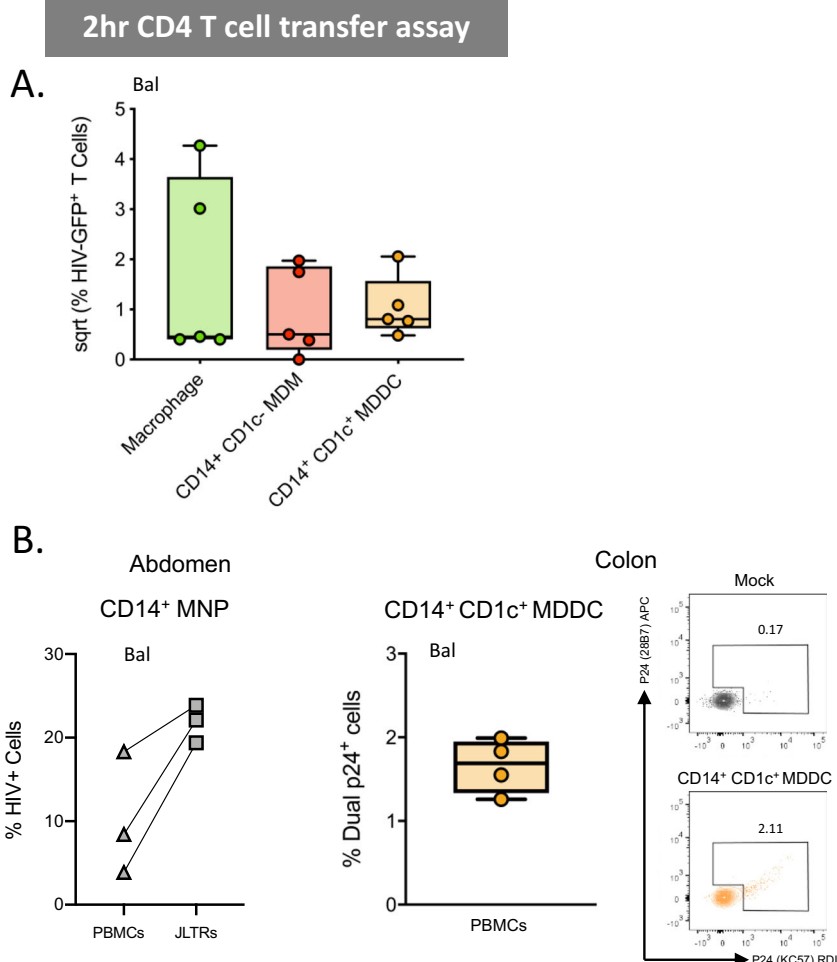

**Fig. 6 1st phase HIV transfer from CD14-expressing mononuclear phagocytes to CD4+ T cells. A** Sorted CD14-expressing dermal cells were incubated with HIV$_{Bal}$ for 2 h and then thoroughly washed off. JLTR cells were added to MNPs at a ratio of 4:1 and cultured for a further 96 h in human fibroblast conditioned media. Transfer of HIV to T cells was assessed using flow cytometry to plot the square root percent of GFP+ T cells, with each dot representing an individual donor ($n = 5$). **B** Left: CD14-expressing MNPs were sorted and 2-h transfer ability to JLTR cells compared to primary activated CD4 T cells derived from PBMCs was assessed ($n = 3$). Primary CD4 T cells were cultured at a ratio of 2T cells: 1 MNP. The number of infected T cells was assessed by flow cytometry using Live/dead NIR and intracellular P24 staining. Three individual MNP donors were plotted with squares representing transfer to primary CD4 T cells and triangles representing transfer to JLTR cells. Right: CD14+ CD1c+ MDDC from four human colonic tissue donors were sorted and 2-h transfer assays to CD4 T cells from PBMCs were setup as above, plotted as box and whisker plots, box representing the upper and lower quartile, central line representing the median, the whiskers the minimum and maximum and each donor represented by an individual dot ($n = 4$). Representative contour plots from one individual donor are shown. Macrophages (green), CD14+ CD1c− MDM (red) and CD14+ CD1c+ (orange).

CD14-expressing cell subsets by flow cytometry (as we have done previously with other ex vivo derived skin MNPs[13,21]) as only small cell yields could be derived for each specific cell subsets (particularly ex vivo MDDCs), even from very large pieces of abdominal skin, and too few live cells could be detected after 96 h of culture. Corresponding with their higher levels of HIV infection, we finally showed that ex vivo MDDCs were the most efficient CD14-expressing cells at transferring HIV to CD4 T cells (Fig. 7E).

**Langerin+ cDC2 are enriched in anogenital tissues and the most efficient cells at HIV uptake and infection**. We next turned our attention to sub-epithelial cDC2 which have been understudied in HIV transmission. We have recently shown that these cells exist in the epidermis of human anogenital cutaneous tissues where they preferentially interact with HIV and transmit it to T cells[21]. Interestingly, we found that, similar to epidermis, langerin+ sub-epithelial cDC2 were significantly enriched in anogenital tissues compared to abdominal skin (Fig. 2B). We also

found that within 2 h these cells took up much more HIV than their langerin− counterparts and cDC1 using both a lab-adapted and transmitted founder HIV strain (Fig. 8A) and could be visualised interacting with the virus in situ using RNAscope following topical infection of foreskin explants (Fig. 8B). Correspondingly, they also transferred HIV to CD4 cells more efficiently at 2 h than langerin− cDC2 (Fig. 8C). Interestingly, although langerin+ cDC2 did not express higher levels of CCR5 (Fig. 9A) or CD4 (Fig. 9B) than their langerin− counterparts, they nevertheless produced higher levels of both HIV lab-adapted BaL strain and a transmitted funder strain after 96 h indicating they supported higher levels of infection (Fig. 9C) and we were able to block this effect with AZT (Fig. 9D). We attempted to block HIV uptake and transfer to CD4 cells using a langerin antibody, as we have done previously with LCs[13], and also to carry out a 96 h MNP-T cell transfer assay but unfortunately these experiments proved impossible due to very small numbers of langerin-expressing sub-epithelial cells that are able to be extracted from tissue.

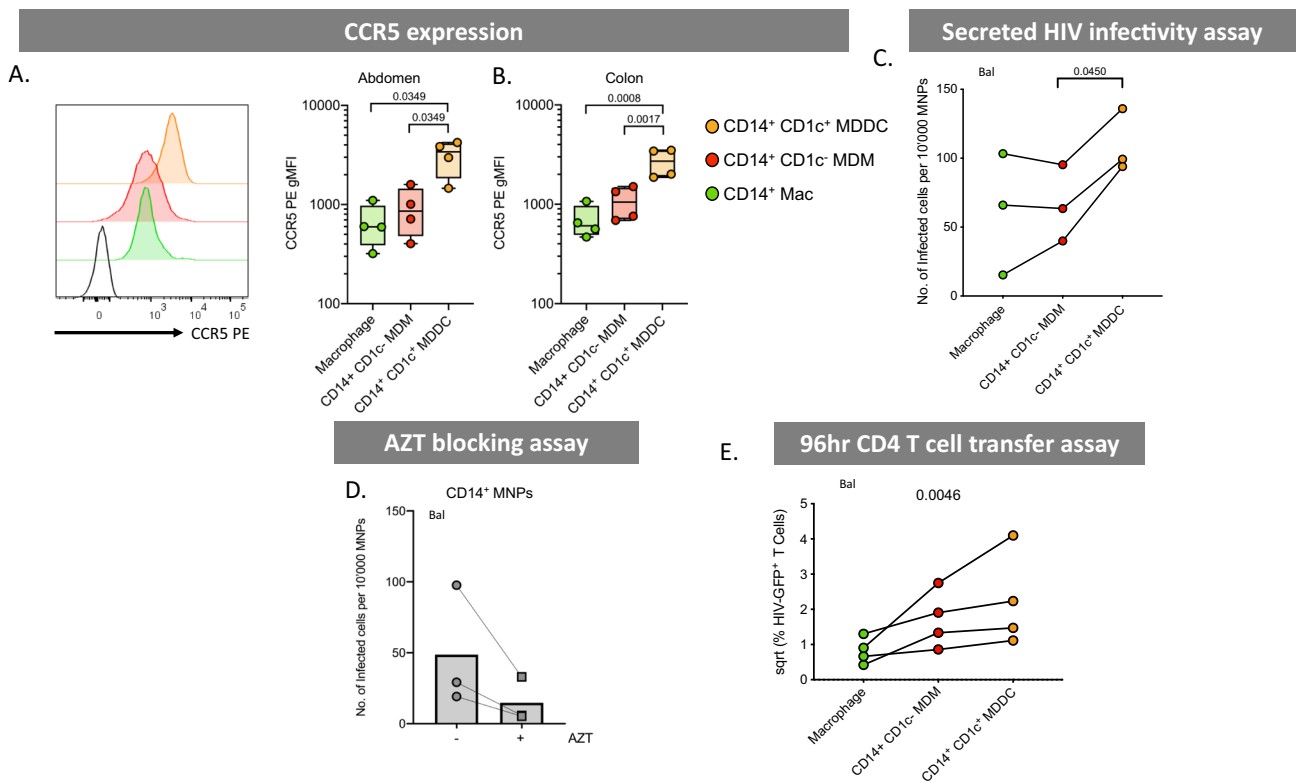

**Fig. 7 HIV infectability and transfer capacity of CD14-expressing dermal mononuclear phagocytes. A** CCR5 expression was determined on CD14-expressing MNPs liberated from collagenase type IV digested tissue. Left: histogram representing one abdominal skin donor with black, unfilled histogram representing FMO control. Right: geometric mean fluorescent intensity (gMFI) minus FMO control plotted from abdominal skin ($n = 4$), plotted as box and whisker plots, box representing the upper and lower quartile, central line representing the median, the whiskers the minimum and maximum of each cell subset and each donor represented by an individual dot. Statistics were generated using a RM one-way ANOVA with Holm–Sidak's multiple comparisons with adjusted P values shown. **B** CCR5 expression was investigated in colon ($n = 4$) plotted as box and whisker plots, box representing the upper and lower quartile, central line representing the median, the whiskers the minimum and maximum of each cell subset and each donor represented by an individual dot. Statistics were generated as in **A**. **C** CD14-expressing MNP subsets were FACS isolated and incubated with HIV_bal for 2 h before being thoroughly washed off. Cells were then cultured for an additional 96 h in human skin fibroblast conditioned media. Cell supernatants were taken and assessed for secreted HIV using a TZMBL infection assay, with the number of infected cells per 10,000 MNPs calculated and graphed. Individual dots represent three donors matched by connecting lines. Statistics were generated as in **A**. **D** Combined CD14-expressing MNPs were sorted and pre-treated for 1 h with 50 μM azidothymidine (AZT) before 2 h culture with HIV_Bal. Cells were washed three times and incubated for a further 48 h with AZT and then washed three more times. Cells were cultured for another 48 h before cell supernatants were collected and secreted HIV was assessed using a TZMBL infection assay. Each donor represented by individual point matched with joining lines, circles represent untreated cells and squares represent AZT treated cells, column representing the mean. **E** JLTRs were added to cell cultures from **C** after supernatants removed, at a ratio of 4:1 and cultured for a further 96 h. Transfer of HIV to T cells was determined using flow cytometry to assess the percent of GFP+ JLTRs. Raw data was square root normalised and plotted, with each dot representing four individual donors with connecting lines matching donors. Statistics were generated using a Friedman test. Macrophages (green), CD14+ CD1c− MDM (red) and CD14+ CD1c+ (orange).

## Discussion

In this study we have investigated the role that sub-epithelial MNPs play in HIV transmission. We began by thoroughly defining the relative proportions MNPs that are present in all human anogenital and colorectal tissues which are the sites of HIV transmission. In doing so we revealed that CD14-expressing cells predominate in these tissues in contrast to abdominal skin. CD14-expressing cells were found in greater proportions in internal mucosal tissues compared to external genital skin. We identified two MNPs subsets that may play a dominant role in HIV transmission; ex vivo CD14+CD1c+ MDDCs and langerin+ cDC2, both of which were present in higher proportions in human anogenital and colorectal tissues compared to abdominal skin.

Tissue macrophages express CD14 and have been classically defined in tissue by their high levels of autofluorescence[26,52]. Non-autofluorescent tissue CD14+ cells have traditionally been referred to as CD14 DCs which have been well characterised in

their ability to bind to and capture HIV and transmit it to CD4 T cells in model systems[11,17,53–56] as well as in intestinal tissue[32–34] and more recently in cervical tissue[15]. The ability of these cells to transmit the virus to CD4 T cells has been assumed to be associated with the potent antigen presenting function of DCs. However, transcriptional profiling has recently led to non-autofluorescent CD14+ CD1c− cells in skin to be redefined as in vivo monocyte-derived macrophages[40]. This is puzzling as macrophages are thought to be weak antigen presenting cells to naïve T cells, especially when compared to DCs and they do not migrate out of tissue to lymph nodes[52]. As ex vivo CD14+CD1c+ MDDCs have been described in inflamed tissue[30,31,37] we therefore carried out transcriptional profiling and found that, in agreement with the literature, CD14+CD1c− cells transcriptionally aligned very closely with macrophages[40] and that CD14+CD1c+ cells aligned more closely with DCs[37]. Adding weight to the hypothesis that these cells were DC-like, we showed that CD14+CD1c+ cells morphologically resembled DCs and,

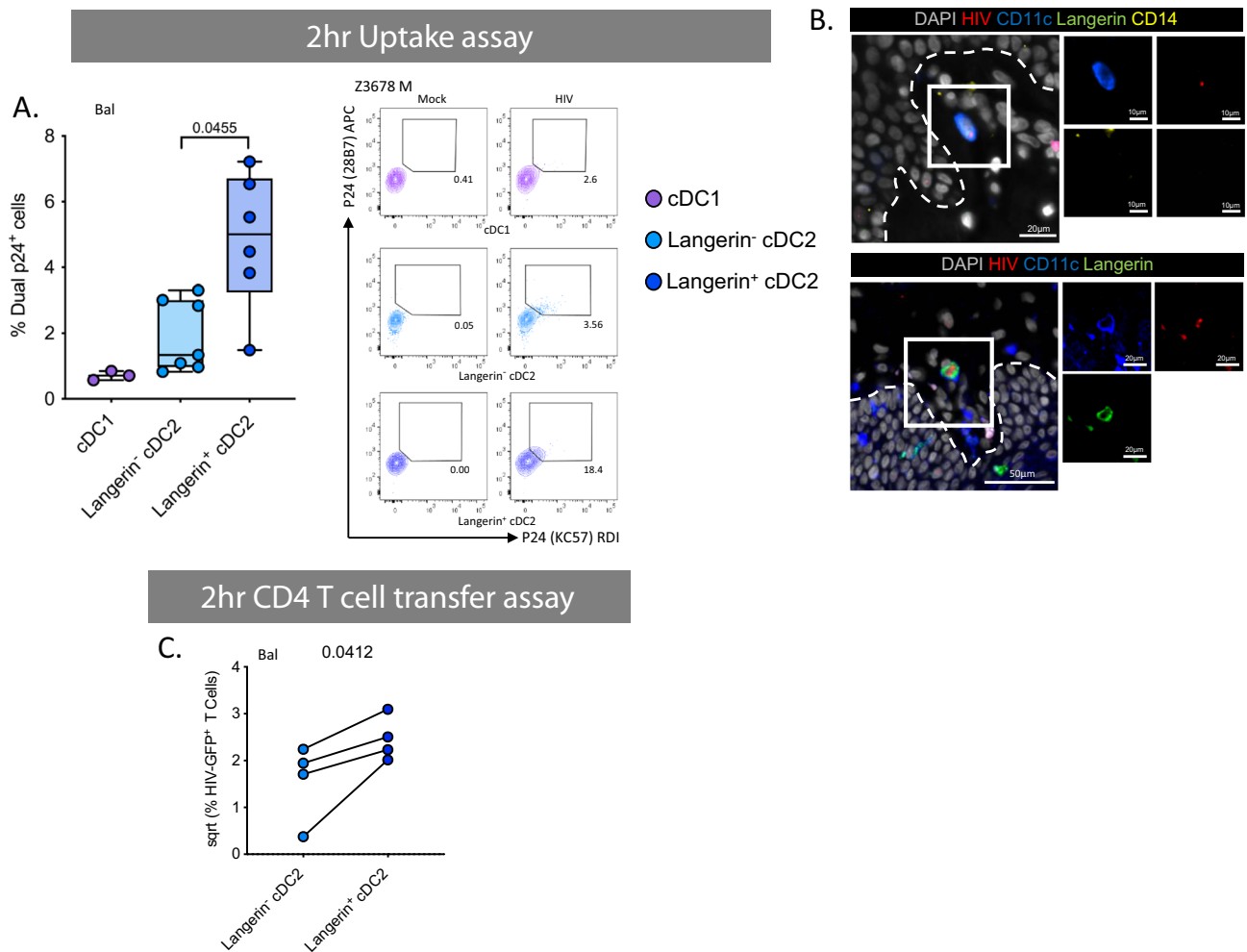

**Fig. 8 HIV uptake and 1st phase transfer capacity of dermal dendritic cells. A** Enzymatically liberated cells from abdominal skin were FACS isolated to obtain a CD45$^+$HLA-DR$^+$ population and cultured with HIV$_{Bal}$ or HIV$_{Z3678M}$ or Mock treated for 2 h before being washed off three times in PBS. Cells were stained for surface markers and two p24 clones, before analysis by flow cytometry. Left: HIV$_{Bal}$ treated cells percentage of dual p24$^+$ cells plotted as box and whisker plots, box representing the upper and lower quartile, central line representing the median, the whiskers the minimum and maximum of each cell subset and each donor represented by an individual dot ($n = 7$, $n = 3$ for cDC1 due to low cell numbers). Statistics were generated using a mixed effects analysis with two-tailed Tukey's multiple comparisons with adjusted $P$ values shown. Right: representative contour plots for HIV$_{Z3678M}$ gating on dual p24$^+$ cells. **B** Human inner foreskin explants were treated with HIV for 2–3 h, before being fixed, paraffin embedded and sectioned. HIV was visualised using RNAscope alongside immunofluoerescence for CD11c, CD14 and langerin as well as DAPI to identify Langerin$^-$ cDC2 (top) and langerin$^+$ cDC2 (bottom). Representative image from $n = 3$ donors. **C** FACS isolated cDC2 divided by langerin expression were cultured with HIV$_{Bal}$ for 2 h before being washed off three times. JLTR cells (CD4 T cells with GFP under the HIV promoter) were then co-cultured for 96 h in fibroblast conditioned media. Flow cytometry was used to analyse the percent of GFP$^+$ T cells. Data was square root normalised and graphed with each donor represented by an individual dot and lines connecting matched donors ($n = 4$). Statistics were generated using a two-tailed paired $T$ test. cDC1 (purple), langerin$^-$ cDC2 (light blue) and langerin$^+$ cDC2 (dark blue).

expressed *CCR7* and spontaneously migrated out of tissue as DCs are known to do, unlike CD14$^+$CD1c$^-$ cells which morphologically resembled macrophages, did not express *CCR7* and remained tissue resident. This migration was enhanced in the presence of the TLR7 agonist imiquimod. These results support the literature which shows that CD1c is an essential marker to define cDC2[57] and inflammatory MDDCs have been shown to express CD14 and CD1c in patients with rheumatoid arthritis and cancer[31,37]. Indeed, it has been recently showed that all human CD1c$^+$ inflammatory DCs derived from ascites are monocyte-derived cells and therefore not cDC1 or cDC2 which are derived from specific bone marrow precursors[30]. However, these cells have not been previously defined in healthy human anogenital tissue which form the portals of HIV entry and we show that they

did differ from inflammatory MDDCs in that they did not express CD1a, CD141 or DC-SIGN.

We next investigated the way CD14-expressing cells interacted with HIV-1$_{BaL}$ and a transmitted/founder clinical HIV-1 strain and found that ex vivo MDDCs cells took up significantly more HIV within 2 h than ex vivo MDMs and autofluorescent macrophages. They also expressed higher levels of the HIV entry receptor CCR5 and correspondingly supported higher levels of productive infection resulting in higher levels of infectious virion secretion and higher levels of transfer of the virus to CD4 T cells. They also expressed lower levels of the HIV restriction factor *APOBEC3G* which could further account for the higher levels of secreted virions by these cells although the HIV Vif protein is known to inhibit the function of this protein. We also investigated

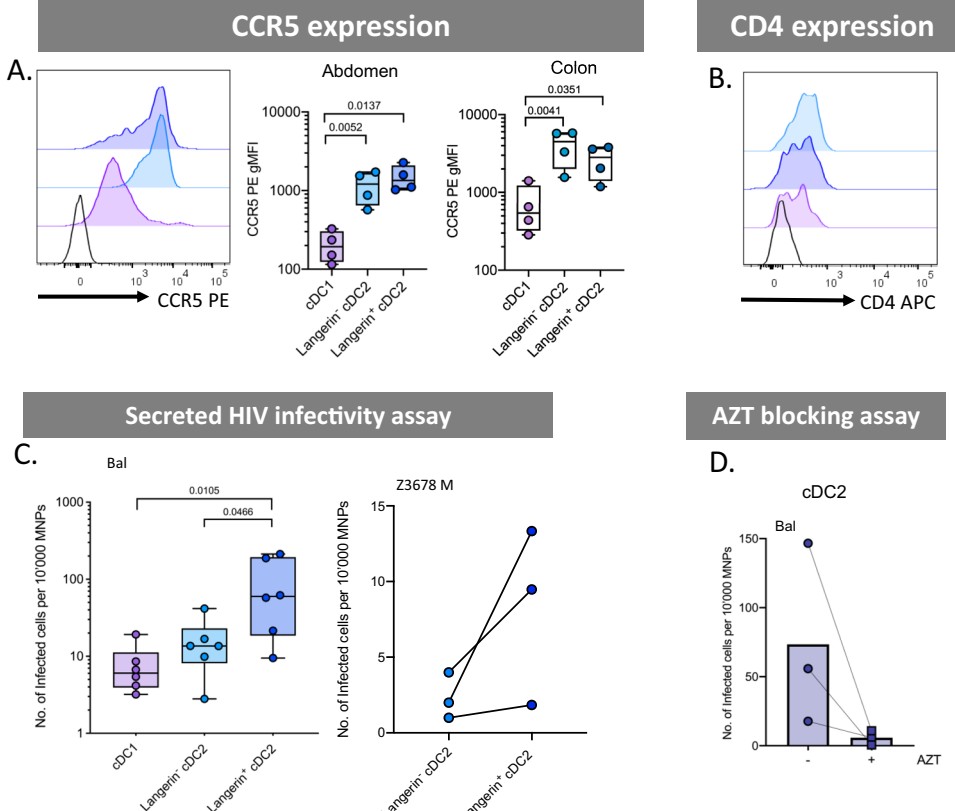

**Fig. 9 HIV infectivity and 2nd phase transfer capacity of dermal dendritic cells. A** HIV co-receptor CCR5 expression was analysed by flow cytometry. Left: representative histogram of one donor, black unfilled histogram represents FMO control. CCR5 gMFI minus the FMO for abdominal skin (middle $n = 4$) and colon (right $n = 4$) plotted as box and whisker plots, box representing the upper and lower quartile, central line representing the median, the whiskers the minimum and maximum of each cell subset and each donor represented by an individual dot. Statistics were generated using a RM one-way ANOVA with Holm–Sidak's multiple comparisons with adjusted $P$ values shown. **B** HIV receptor CD4 expression levels on one abdominal skin donor with isotype control represented by black, unfilled histogram. **C** Cell supernatants from FACS isolated dermal DCs infected with HIV were taken for TZMBL infectivity assays and the number of infected TZMBL cells per $10^4$ MNPs were calculated. Left: $HIV_{Bal}$ infectivity plotted as box and whisker plots as in **A**, $n = 7$. Statistics were generated using a RM one-way ANOVA with Holms–Sidak's multiple comparisons with adjusted $P$ values shown. Right: $HIV_{Z3678M}$ infectivity assay with each dot representing individual donors matched by connecting lines ($n = 3$). Due to low cell numbers cDC1 data could not be obtained. **D** Combined sorted cDC2 were pre-treated with or without azidothymidine (AZT) before 2-h $HIV_{Bal}$ infection. AZT was washed off after 48 h and cell supernatants collected after a further 48 h. Secreted HIV was assessed using a TZMBL infection assay. Each donor represented by individual point matched with joining lines, circles represent untreated cells and squares represent AZT treated cells, column representing the mean. cDC1 (purple), langerin⁻ cDC2 (light blue) and langerin⁺ cDC2 (dark blue).

the expression of the key HIV restriction factor SAMHD1 and found that it was expressed at roughly equal levels across MNP subsets except cDC1 which expressed higher levels of this protein. It is of note that the SAMHD1 antibody clone we used did not distinguish whether the SAMHD1 was phosphorylated (no restriction) or not (antiviral). Uptake of HIV by MNPs at early time points is mediated by lectin receptors and previously DC-SIGN has shown to be key a lectin receptor involved in HIV uptake by CD14-expressing cells and has also been implicated in transfer to CD4 T cells[11,29,32,33,45]. Similarly Siglec-1 has also been implicated[15,47]. However, ex vivo MDDCs did not express DC-SIGN so this receptor cannot be responsible for the efficient early uptake observed at 2 h by these cells. This is an important observation as soluble DC-SIGN designed to block HIV interacting with MNPs has been proven ineffective in blocking HIV transmission[58]. Ex vivo MDDCs did however express Siglec-1, but at lower levels than ex vivo MDMs and autofluorescent macrophages which expressed the highest Siglec-1 levels. We therefore investigated the role of Siglec-1 in early uptake and found that in all CD14-expressing cell types, those cells that contained HIV expressed higher levels of Siglec-1 than uninfected

cells. Furthermore, a Siglec-1 blocking antibody was able to partially block HIV uptake which corresponded with the levels of Siglec-1 surface expression. Therefore, the mechanism which allows ex vivo MDDCs to take up HIV more efficiently than other CD14-expressing cells remains to be elucidated and will be the subject of a future study. However, other than DC-SIGN we did not detect any difference in the expression profiles of known HIV binding CLRs between ex vivo MDDCs and other CD14⁺ cells. Ex vivo MDDCs did however express higher levels of CLEC5A which opens an avenue of investigation. It is of note that ex vivo CD14⁺ tissue resident macrophages and MDMs were able to take up HIV (albeit significantly less so than MDDCs). The possibility therefore exists that these cells act as virus reservoirs that transmit viral particles to DCs which then migrate and transmit the virus to T cells.

Despite their importance in antigen presentation, cDC2 have been understudied in HIV transmission with almost all studies focussing on langerin-expressing LCs or DC-SIGN expressing CD14⁺ cells[10,59]. Recently we reported that cDC2-like cells were present in the epidermis of anogenital tissues where they predominated over LCs and preferentially become infected with HIV

and transmitted the virus to CD4 T cells[21]. Pena-Cruz and colleagues made similar observations in vaginal epithelium[22]. Importantly, we showed that in anogenital tissues the majority of these cells expressed langerin. We therefore investigated the role of lamina propria cDC2 in HIV transmission. We noticed that a greater proportion of sub-epithelial cDC2 in anogenital and colorectal tissues expressed langerin than in abdominal dermis. This was especially the case in the inner foreskin, penile urethra, vagina and rectum. Importantly, we found that langerin-expressing cDC2 were much more efficient at HIV uptake after 2 h than their non-langerin-expressing counterparts and were correspondingly much more efficient at transferring the virus to CD4 T cells at the same early time point. In fact, these cells were the most efficient of all MNPs at transfer of HIV to CD4 T cells, meaning despite their relatively lower frequency compared to other MNP cell types, they are nevertheless likely to be key players in HIV transmission. Furthermore, despite the fact that no differences in surface CCR5 expression were detected between the two cells types, langerin-expressing cDC2 also secreted higher levels of infectious HIV virions after 96 h using both the lab-adapted Bal strain and a clinical transmitted/founder strain. Other than langerin expression these two cell types expressed an identical array of surface lectin receptors so we hypothesise that langerin must be mediating these effects. In LCs langerin is well known to bind HIV and mediate efficient HIV uptake[13,59–61] so we hypothesise here that these cells efficiently bind HIV via langerin which concentrates HIV on the cells surface allowing for greater HIV-CCR5 binding and infection as well as endocytic uptake. We made multiple attempts to test this hypothesis using a langerin blocking antibody as we did previously with LCs[13] but unfortunately these experiments proved impossible using the very small numbers of this cell type we could extract from abdominal or genital skin or mucosal tissue.

It is of note that under the condition used in this study, virus spread from T cells to T cells may occur. It is possible that the efficiency of T cell-to-T cell spread may be differentially altered in the presence of different MNPs (e.g. via cytokine release). We believe this is unlikely to be substantial especially in the context of early phase (2 h) transfer which is from MNP intracellular caves which is unlikely to be a powerful inducer of cytokine release. However, if this mechanism does contribute it clearly varies amongst MNP subsets in targeting spread to CD4 T cells and therefore it is no less relevant to transmission.

A clear strength of this study is that it has been exclusively conducted using human tissues including the anogenital and colorectal tissues that HIV may encounter during sexual transmission. However, these kinds of experiments also come with limitations. They are very laborious and time consuming and only a very small number of cells can be isolated for each specific subset, especially ex vivo MDDCs and even more so for langerin[+] cDC2 which were both a key focus of this study. This severely limits the parameters that can be included in our assays and many experiments proved impossible such as blocking assays using langerin[+] cDC2. Furthermore, these cells are very difficult to culture once isolated from tissue and despite our optimised cell extraction and ex vivo culture protocols[39] functional assays requiring 96 h of culture were extremely difficult to perform. We did manage to perform HIV infection assays using infected MNP culture supernatants but there were too few live cells to gate on to measure direct infection of any MNP cell type by flow cytometry as we have done previously for epidermal cells[13,21] and we were unable to perform transfer assays at 96 h for langerin[+] cDC2. These constraints also meant that we could only repeat a few key observations with a clinical transmitted/founder strain. All uptake assays were confirmed at least once using this strain and also the cDC2 infectivity assays. In all experiments similar trends were

observed with both virus strains. Furthermore, we have previously observed similar trends using the same two strains of HIV in epidermal LCs and CD11c[+] DCs[21].

In conclusion, we have identified two sub-epithelial MNPs that may play a role in transmission of HIV; ex vivo CD14[+]CD1c[+] MDDCs and langerin[+] cDC2. Both were able to preferentially take up the virus within 2 h and support higher levels of HIV infection than other MNP subsets. Previously, many studies have focussed on the role of LCs in sexual transmission of HIV[59] as these were considered most likely to interact with HIV as they are closest to the epithelial surface. We also demonstrated the importance of a second epidermal DC subset, resembling activated cDC2[21]. However, as genital trauma and inflammation are clearly associated with HIV transmission, especially in sub-Saharan Africa[3–5], and current PrEP regimens, if available, are not efficient at blocking transmission across an inflamed mucosa[7–9], it is important that the role of sub-epithelial MNPs are also examined as we have done here. This is relevant to vaccine design as, to protect against initial infection, the route of transmission and which local immune defence mechanisms can be harnessed or impaired needs to be understood. The current targets of systemic vaccines as broadly neutralising antibodies, antibody dependent cytotoxicity and systemic CD8 T cells may not be enough. Local immunity in the anogenital mucosa such as resident memory T cells could be induced by mucosal vaccines or maintained by local tissue DCs after initial stimulation by systemic vaccine adjuvants in lymph nodes draining the site of application. Therefore, it is important to determine which specific subsets of MNPs pick up the virus and deliver it to specific subsets of CD4 T cells. Here we have identified two MNP subsets of interest and together with our recently discovered epidermal CD11c[+] DCs[21] and LCs, the next step is to determine with which T cells subsets these cells interact. Finally, defining the interactions between HIV and its mucosal target cells should aid in the development of blocking agents that can be used in modified PrEP regimens to block MNP infection via the anogenital and colorectal mucosa.

## Methods

**Sources of tissues and ethical approval.** This study was approved by the Western Sydney Local Area Health District (WSLHD) Human Research Ethics Committee (HREC); reference number HREC/2013/8/4.4(3777) AU RED HREC/13/WMEAD/232. Healthy human tissue was obtained from a range of plastic surgeons and written consent was obtained from all donors.

**Tissue processing.** MNP were isolated from abdominal tissue using our optimised collagenase-based digestion process[21,39]. Skin was collected immediately after surgery, stretched out and sectioned using a skin graft knife (Swann-Morton, Sheffield, UK) and the resulting skin grafts passed through a skin graft mesher (Zimmer Bionet, Warsaw, IN, USA). The meshed skin was placed in RPMI1640 (Lonza, Switzerland) with 0.14 U/ml dispase (neutral protease, Worthington Industries, Columbus, OH, USA) and 50 µg/mL Gentamicin (Sigma-Aldrich, St Louis, MO, USA) and rotated at 4 °C overnight. The skin was then washed in PBS and dermis and epidermis were mechanically separated using fine forceps. Dermal tissue was cut into 1–2 mm pieces using a scalpel. Dermal and epidermal tissue was then incubated separately in RPMI1640 containing 100 U/ml DNase I (Worthington Industries) and 200 U/ml collagenase Type IV (Worthington) at 37 °C for 120 min in a rotator. The cells were then separated from undigested dermal and epidermal tissue using a tea strainer. The supernatants were then passed through a 100-µm cell strainer (Greiner Bio-One, Monroe, NC, USA) and pelleted. The cell pellet was then passed again through a 100-µm cell strainer and washed twice more in PBS. The epidermal suspension was spun on a Ficoll-Paque PLUS (GE Healthcare Life Sciences, Little Chalfont, United Kingdom) gradient and the immune cells harvested from the Ficoll-PBS interface. Dermal cells were enriched for CD45-expressing cells using CD45 magnetic bead separation (Miltenyi Biotec, San Diego, CA, USA). Cell suspensions were then counted and/or labelled for flow cytometric phenotyping of surface expression markers or for flow sorting. This protocol was modified for anogenital tissues as follows: for skin and type II mucosa (labia, foreskin, glans penis, fossa navicularis, vagina, ectocervix and anal canal), small shallow scalpel cuts were made to the epithelial surface before overnight dispase II treatment; for type I mucosa (endocervix, penile urethra, rectum and colon) no dispase treatment was required and tissue was digested using two

successive 30 min digestions with collagenase type IV. For spontaneous migration assays, following mechanical separation of dermis and epidermis tissue was cultured for 24 h in RPMI 10% FCS, 50 U/ml DNase I and 25 µg/ml gentamicin before cells were collected from culture media. For TLR stimulated experiments culture media contained 1 µg/ml imiquimod (R837, invivoGen). Isolated cells were then treated as above.

**Preparation of in vitro monocyte-derived dendritic cells and macrophages**. CD14+ monocytes were derived from human blood and cultured for 6 days in RPMI with 500 U/ml interleukin (IL)-4 and 300 U/ml granulocyte-macrophage colony-stimulating factor (GM-CSF) to produce in vitro derived MDDC or in human serum to generate in vitro derived MDM as described previously[23,53,62–64].

**Flow cytometry and sorting**. Cells were labelled in aliquots of $1 \times 10^6$ cells per 100 µl of buffer, according to standard protocols. Nonviable cells were excluded by staining with Live/Dead Near-IR dead cell stain kit (Life Technologies, Carlsbad, CA, USA). Flow cytometry was performed on Becton Dickenson (BD, Franklin Lakes, NJ, USA) LSRFortessa, LSRII, CantoII and Symphony flow cytometers with BD DIVA software (V 8.0) and data analysed by FlowJo (Treestar V 10.0). Fluorescence Activated Cell Sorting (FACS) was performed on a BD FACS Influx (100 µm nozzle and 20 pounds/square inch). Sorted cells were collected into FACS tubes containing RPMI culture media supplemented with 10 µM HEPES (Gibco, Waltham, MA, USA), non-essential amino acids (Gibco), 1 mM sodium pyruvate (Gibco), 50 µM 2-mercaptoethanol (Gibco), 10 µg/ml gentamycin (Gibco) and 10% (v/v) FCS (from herein referred to as DC culture media). The antibodies were purchased from BD, Miltenyi, Bio Legend (San Diego, CA, USA), Beckman Coulter (Brea, CA, USA), eBioscience (San Diego, CA, USA), Merck (Kenilworth, NJ, USA) and R&D Systems (Minneapolis, MN, USA) as follows; BD: CD45 BV786 (HI30); 1 µl/100 µl, CD45 PE (HI30) 2 µl/100 µl, HLA-DR BUV395 (G46-6) 0.5 µl/100 µl, CD1a BV510 (HI149) 1.5 µl/100 µl, CD14 BUV737 (M5E2) 2.5 µl/100 µl, CD14 BV421 (M5E2) 2.5 µl/100 µl, CD11c PE CF594 (B-Ly6) 1.5 µl/100 µl, CD1c PE (F10/21A3) 2.5 µl/100 µl, CD141 BV711 (1A4) 2 µl/100 µl, CD33 (Siglec-3) APC (WM53) 2 µl/100 µl, CD161 (Clec5B) APC (DX12) 2 µl/100 µl, CD184 (CXCR4) PE (12G5) 4 µl/100 µl, CD103 PE (Ber-ACT8) 10 µl/100 µl, CD4 APC (RPA-T4) 4 µl/100 µl, CD206 (MR) BUV805 (19.2) 0.5 µl/100 µl, CD209 (DC-SIGN) APC (DCN64) 2.5 µl/100 µl, CD209 (DC-SIGN) PE (DCN46) 2.5 µl/100 µl, CD80 PE (L307.4) 2 µl/100 µl, CD83 APC (HB15e) 2 µl/100 µl, CD86 APC (2331 (FUN-1) 2 µl/100 µl, CD371 (Clec12A) AF647 (50C1) 2 µl/100 µl, Mouse IgG1 APC, Mouse IgG2b APC, Mouse IgG1 PE, Mouse IgG2a PE. Miltenyi: CD14 Vioblue (TUK4) 1.5 µl/100 µl, CD195 (CCR5) PE (REA245) 2.5 µl/100 µl, Clec7A (Dectin-1; CD369) PE (REA515) 2 µl/100 µl, Clec9A (CD370) PE (8F9) 5 µl/100 µl, CD1c (BDCA1) PE-Vio770 (AD5-8E7) 2.5 µl/100 µl, CD141 APC (AD4-14H12) 2 µl/100 µl, CD207 (Langerin; Clec4K) Vioblue (MB22-9F5) 1.5 µl/100 µl, HLA DR PerCP (AC122) 2 µl/100 µl, langerin PE-Vio770 (MB22-9F2) 1.5 µl/100 µl, Mouse IgG2a PE and Mouse IgG1 APC. Bio Legend: CD169 (Siglec-1) PE (7-239) 1 µl/100 µl, DEC205 PE (HD30) 1 µl/100 µl, XCR1 APC (S15046E) 5 µl/100 µl and CD172a (SIRPα) APC/Fire750 (SE5A5) 3 µl/100 µl. Beckman Coulter: CD207 PE (DCGM4) 1 µl/100 µl, CD206 (MR) PE (3.29B1.10) 2 µl/100 µl and Mouse IgG1 PE (679.1Mc7). eBioscience: CD91 eFluor660 (A2MR-a2) 1 µl/100 µl, CD169 (Siglec-1) PE eFluor610 (7-239) 2.5 µl/100 µl, CD172a (SIRPa) PeCP-eFluor710 6 µl/100 µl and Mouse IgG1 eFluor660. Merck: SAMHD1 (I-19-18) 1 µl/100 µl. R&D Systems: CD299 (L-SIGN; DC-SIGNR) PE (120604) 2 µl/100 µl, CD367 (Clec4A; DCIR) PE (216110) 4 µl/100 µl, CD368 (Clec4D) PE (413512) 2 µl/100 µl, Clec5A APC (283834) 3 µl/100 µl, Clec5C APC (239127) 2 µl/100 µl, Clec6A (Dectin-2) APC (545943) 2 µl/100 µl, CD301 (Clec10A) PE (744812) 2 µl/100 µl, Clec14A APC (743940) 2 µl/100 µl, Siglec-9 APC (191240) 2 µl/100 µl, Siglec-16 APC (706022) 5 µl/100 µl, Mouse IgG1 PE, Mouse IgG2b PE, Mouse IgG2b APC and Mouse IgG2a APC. For HIV detection cells were incubated with Cytofix/Cytoperm solution (BD) for 15 min at room temp and washed with permeability wash (1% BSA, 0.05% Saponin, 0.1% NaN₃ in PBS). Samples were then blocked with 10% HuAb serum for 10 min and stained with two antibody clones to HIV p24, KC57 PE (Beckman Coulter) 1 µl/100 µl and 28B7 APC (Medimabs, Canada) 1 µl/100 µl.

**RNA seq**. Individual MNP subsets from human abdominal dermis and epidermis were liberated and FACS isolated (Supplementary Fig. 1A, B). Total RNA (1 ng in 2 µl) was reverse transcribed and amplified using the Smart-seq2 protocol of Picelli et al.[65,66]. Transcripts were first mixed with 1 µl of 10 µM anchored oligo-dT primer (5′-AAGCAGTGGTATCAACGCAGAGTACT₃₀VN-3′), and 1 µl dNTP mix (10 mM) and heated to 72 °C for 3 min then placed on ice immediately. A first strand reaction mix containing Superscript II, TSO primer (5′-AAGCAGTGGT ATCAACGCAGAGTACATrGrG+G-3′), 1 µM, 1 M betaine, 6 mM MgCl₂ 5 mM DTT, RNaseOUT (10U) and 2 µl 5X first strand buffer, was added to each sample. Transcripts were reverse transcribed by incubating at 42 °C for 60 min, followed by 10 cycles of 50 °C, 2 min and 42 °C, 2 min. The resulting cDNAs were subsequently amplified by adding 12.5 µl KAPA HiFi 2X HotStart ReadyMix, 10 µM ISPCR primer (0.25 µl, 5′-AAGCAGTGGTATCAACGCAGAGT-3′) and 2.25 µl H₂O and heating to 98 °C, 3 min, followed by 15 cycles of 98 °C, 20 s, 67 °C, 15 s, 72 °C, 6 min. After a final incubation at 72 °C, 5 min, samples were held at 4 °C until purification. Samples were

purified by addition of Agencourt AMPure XP beads (1:1 v/v), 8 min, room temperature, followed by magnetic bead capture and two washes with 200 µl 80% ethanol. Beads were dried and cDNAs were eluted into 17.5 µl elution buffer EB. Products were checked on Agilent Bioanalyser High Sensitivity DNA chips and yield was estimated by Quant-iT PicoGreen dsDNA Assay. Sequencing libraries were prepared from 2 ng product using Nextera XT reagents in accordance with the supplier's protocol except that PCR was limited to eight cycles to minimise read duplicates, and final library elution was performed using 17.5 µl elution buffer EB. Libraries were pooled and sequenced on the Illumina HiSeq 2500 platform generating 50 base pair single-end sequences. Raw sequence data was aligned to the UCSC human reference genome (hg19) using the TopHat2 software package[67]. Aligned sequencing reads were summarised to counts per gene using the Read Assignment via Expectation Maximisation (RAEM) procedure[68] and reads per kilobase per million mapped reads (RPKM) values were calculated in SAMMate (v 2.6.1)[68].

**HIV transfer assay**. Ex vivo-derived dermal mononuclear phagocyte (MNP) subsets were tested for their ability to transfer HIV to activated T cells as described previously for Epidermal MNPs[13,2,1]. Cells were liberated from abdominal skin as described above and individual subsets were FACS isolated (Supplementary Fig. 1A). A minimum of $1 \times 10^4$ and $3 \times 10^4$ cells were cultured with HIV_Bal for 1st Phase and 2nd phase transfer assays respectively, at an MOI of 1 for 2 h before being washed off three times with PBS. For 1st Phase transfer assays, JLTR cells (CD4 T cells which express GFP under the HIV promoter) were co-cultured at a ratio of 4:1 (T cells:MNPs) or activated CD4+ PBMCs at a ratio of 2:1, for 96 h in DC culture media for 96 h in DC culture media. For 2nd phase transfer assay, cells were cultured for 96 h before addition of JLTR cells. Flow cytometry was used to determine the percent of GFP+ JLTRs or p24+ PBMCs.

**Secreted HIV infectivity assay**. Cell supernatants from 2nd phase transfer assays were taken prior to the addition of JLTRs (96 h after virus had been washed off). Supernatants were cultured with TZMBL cells (HeLa cell derivatives expressing high levels of CD4 and CCR5 and containing the β-galactosidase reporter gene under the HIV promoter) for a further 72 h on 96-well flat-bottom plates. Supernatants were removed and TZMBL development solution added (0.5 M potassium ferricyanide, 0.5 M potassium cyanide and 50 mg/ml X-gal) and incubated for 1 h to detect LTR β-galactosidase reporter gene expression. Development solution was removed, and cells fixed with 4% (v/v) PFA (diluted in PBS) for 15 min. Infectivity was quantified using an EliSpot plate reader (AID, Strasberg, Germany). For AZT blocking assays, cells were pre-treated with 50 µM AZT (National Institutes of Health AIDS reagent programme) for 1 h at 37 °C before addition of HIV. At 48 h AZT was thoroughly washed off and the assay continued on as described above.

**HIV uptake assay**. Ex vivo dermal MNP subsets were tested for their ability to take up HIV virions. Cells were liberated from abdominal tissue through enzymatic digestion and live HLA-DR+ cells were FACS isolated (Supplementary Fig. 1C). Sorted cells were cultured for 2 h with HIV_Bal or HIV_Z3678M at an MOI of 3 before being washed of 3x with PBS. Cells were labelled for flow cytometric analysis as described above and analysed to determine the percent of dual p24+ cells. For Siglec-1 blocking assays, cells were pre-treated with 20 µg/ml Siglec-1 mAb (Invitrogen, HSn 7D2) for 30 min before HIV was added.

**HIV explants on human penile tissue**. Inner foreskin and penile urethra explants were infected with HIV_Bal as previously described[21]. Briefly, small scalpel cuts were punctured into the mucosal surface of the tissue before cloning cylinders (8 × 8 mm, Sigma-Aldrich) were adhered to the tissue using surgical glue (B Braun, Germany). Explants were cut around the cloning cylinder and placed onto gel foam sponge (Pfizer, New York, NY, USA) soaking in DC culture media in a 24-well plate. HIV was diluted in 100 µl of PBS at a TCID₅₀ of 3500 and then added to cloning cylinders. In all, 100 µl of PBS was added to mock samples to prevent tissue drying out. Explants were cultured for 2–3 h before virus/PBS was removed and cylinders washed out 3x with PBS. After removal of cloning cylinders, tissue explants were placed in 4% PFA (Electron Microscopy Sciences, Hatfield, PA, USA) for 24 h before paraffin embedding.

**RNAscope and immunofluorescent staining of tissue**. All microscopy staining was carried out on 4 µm paraffin sections. Sections were baked at 60 °C for 45 min, dewaxed in xylene followed by 100% ethanol and dried. Detection of HIV RNA was achieved using the RNAscope 2.5HD Reagent Kit-RED (Cat: 322360, ACD Bio, Newark, CA, USA) as previously described[21]. Sections underwent antigen retrieval in RNAscope target Retrieval buffer (RNAscope Kit) at 95 °C for 20 min in a decloaking chamber (Biocare, Pacheco, CA, USA), airdried and sections encircled with a hydrophobic pen. Unless stated otherwise slides were washed 2x for a total of 5 min. Slides were air dried and treated with Bloxall (Vector Laboratories, SP-6000, Burlingame, CA, USA) for 10 min, washed in Milli-Q water and then incubated with protease pre-treatment-3 (RNAscope kit) for 20 min at 40 °C. Slides were washed in Milli-Q water and incubated with custom-made HIV-1_Bal probe (REFL 486631, ACD Bio) for 2 h at 40 °C. Slides were washed in RNAscope Wash Buffer (RNAscope Kit), followed by sequential addition of amplification reagents 1–6 (RNAscope Kit) as follows with washes in RNAscope wash buffer between

each amp: Amp 1: 30 min 40°, Amp 2: 15 min 40°, Amp 3: 30 min 40 °C, Amp 4: 15 min 40 °C, Amp 5: 30 min room temperature and Amp 6: 15 min room temperature. Fast Red chromagen solution was added to sections at a dilution of 1:60 Red-B:Red-A (RNAscope Kit), at room temperature for 5 min. Slides were washed firstly in Milli-Q water followed by 2x TBS (Amresco, Cat: 0788) for a total of 5 min. Slides were then blocked for 30 min (0.1% (w/v) saponin, 1% (w/v) BSA and 10% (v/v) donkey serum) in preparation for immunofluorescence staining. From herein, between each step slides were washed in TBS unless otherwise stated. The first round of primary antibodies were incubated for 1 h at room temperature, these included goat anti-human langerin (AF2088; 1:500) and rabbit anti-human CD14 (ab133503; 1:250). Secondary antibodies were then added for 30 min at room temperature, these included donkey anti goat-488 and donkey anti-rabbit-755 (Molecular Probes; 1:400). Slides were blocked again as above with the addition of 10% (v/v) rabbit serum, followed by 0.5% (v/v) PFA for 15 min. Rabbit anti-human CD11c (ab216655) underwent buffer exchange using 50 kDa amicon filters (Merck) so as to place antibodies in 0.1 M NaHCO3, balanced to a pH of 8.3–8.5. This was followed by conjugation to sulfo-Cyanine5 using the lumiprobe sulfo-Cyanine5 antibody labelling kit (3321-10rxn, Lumiprobe, Hallandale Beach, FL, USA), following the manufacturer's instructions. The conjugated rabbit CD11c-Cy5 was added to sections and incubated at 4 °C overnight (1:50). Slides were mounted with prolong SlowFade Diamond Antifade with DAPI (Molecular Probes, Eugene, OR, USA, S36973). Sections were imaged on the Olympus VS120 slidescanner using channels FITC, TRITC, Cy5 and Cy7 using Olympus-asw software (version 2.9), deconvolved using Huygens Professional (20.04) and analysed using Olyvia (version 2.9) and Fiji (Madison version).

**Statistical analysis.** For comparisons on unmatched donor data Kruskal–Wallis with Dunn's multiple comparison tests were performed to correct for the multiple contrasts made. Repeated measures one-way ANOVA with a Holm–Sidak's multiple comparisons tests were used for experiments comparing more than two groups with donor matched experiments with equal number of data points. For experiments with unequal numbers of donor matched data points mixed effects analysis with Tukey's multiple comparisons were performed. All multiple comparisons calculated adjusted $P$ values. Normality was assessed using an Anderson Darling test. Spearman's tests were used to assess heteroscedasticity. All statistics were performed using GraphPad Prism Version 8.4.3.

**Reporting summary.** Further information on research design is available in the Nature Research Reporting Summary linked to this article.

## Data availability

The data presented in this study are available from the corresponding author upon reasonable request. Source data used to generate figures are provided in the source data file. The RNA sequencing data in Fig. 2 have been deposited in the Gene Expression Omnibus (GEO) database under accession code: GSE166639. Source data are provided with this paper.

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

## Acknowledgements

All flow cytometry, FACS and cell imaging were performed at the Westmead Flow Cytometry Core Facility and Cell Imaging Core Facility supported by the Westmead Institute, Westmead Research Hub, Cancer Institute New South Wales and National Health and Medical Research Council.

## Author contributions

J.W.R. guided all the experiments in the manuscript. R.A.B. helped guide key experiments to begin the project and provided extensive intellectual input throughout. K.M.B. assisted and gave intellectual input to all experiments. E.E.V. assisted with colonic tissue processing and analysis. H.R. assisted with tissue processing and HIV functional assays. H.B. gave extensive intellectual input for microscopy experiments. T.R.O. helped with tissue processing and MNP quantification by microscopy. A.A. performed the TLR stimulated cell migration experiments. J.D.G. guided the RNAseq. P.V. and G.P.P. conducted the RNAseq analysis. J.F. conducted the Giemsa staining. N.N. provided intellectual input into the infection and transfer assay experiments. J.K.L., L.B., P.H., M.P.G., A.DR., F.R., G.C., G.J.J. and A.J.B. provided human tissue specimens and intellectual input. E.P. assisted with statistical analysis. E.H. provided the clinical transmitted founder isolate. S.N.B., M.A.H. and A.L.C. provided significant intellectual input. A.N.H. conceived of and guided the study.

## Competing interests

The authors declare no competing interests.
