## [Peer Review File · Nature Communications]

Reviewers' Comments:

Reviewer #1:

Remarks to the Author:

In this study, Rhodes et al. described properties of mononuclear phagocytes (MNPs) in human anogenital and colorectal tissues with a particular focus on C type lectin and other receptors that are of importance to HIV-1 spread. The authors also examined how specific subsets of MNPs interact with HIV-1 and potentially deliver the virus to CD4+ T cells. They observed that two MNP subsets, CD14+CD1c+CD11c+ monocyte-derived dendritic cells and langerin-expressing dendritic cells 2 (DC2), take up HIV-1, get infected, and transfer the virus to T cells more efficiently than other subsets. The information described in this manuscript is certainly useful to HIV researchers, especially those who are interested in mucosal infection. However, there are several shortcomings including the incremental nature of the advance relative to their previous Nature Communications paper, technical limitations with regard to the functional assays, and insufficient quality of the imaging data. These and other weaknesses are described below.

- 1) While the comprehensive description of subepithelial MNP subsets with regard to cell surface receptors and the ability to deliver virus to T cells is appreciated, the study does not represent a fundamentally new finding/concept. Rather, this is an extension of the previous work on epidermal MNPs in anogenital and colorectal tissues from this laboratory (reported in Nature Communications last year) to another parts of the same tissues.
- 2) Fig 4B is not convincing. In top right panel, all HIV signals are associated with CD14-negative but not CD14-positive cells. In bottom right CD14+CD1c+ cells are shown, but the level of HIV associated with these cells are very low compared to those observed in top right, contradicting the outcomes of Fig 4A, which suggest that CD14+CD1c- cells are most efficient in virus uptake.
- 3) In the Fig 4F and 5D experiments, the authors cocultured the GFP reporter T cell line (JLTR) with infected MNPs for 96 h and used the GFP-positive cell number as the readout for the ability of MNPs to transfer HIV to CD4+ T cells. There are three concerns with this set of experiments.
 - a) The ability of virus transfer to T cells is affected not only by the ability of MNPs to capture/propagate HIV-1 but also by T cell adhesion molecules that bind MNP surface molecules. Since T cell lines may not fully recapitulate surface protein profiles of primary T cells, to test the true ability of MNPs to deliver HIV-1 to "T cells", the coculture experiments should be repeated using primary T cells.
 - b) Under the condition used in this study, virus spread from T cells to T cells can occur. The efficiency of T cell-to-T cell spread can be differentially altered in the presence of different MNPs (directly or indirectly, e.g., via cytokine release). Therefore, the difference seen between MNP subsets in this assay may not necessarily represent the difference in efficiency of MNP-to-T HIV transfer.
 - c) Due to the technical limitations described in Discussion, the surface receptor involved in HIV transfer could not be identified. Therefore, the mechanism by which specific subsets of MNPs promote infection in JLTRs remains unknown.

Reviewer #2:

Remarks to the Author:

Rhodes and colleagues have expanded their previous findings on epithelial mononuclear phagocytes (MNPs) and studied their roles in HIV-1 transmission. They have isolated MNPs from various human anogenital and colorectal sub-epithelial tissues and classified them into 6 subsets of MNPs. They show that epidermal MNPs, in particular CD1c+ monocyte-derived DCs and langerin+ cDC2, uptake HIV-1, leading to cis infection of MNPs and trans-infection of CD4+ T cells. They have tested a role of lectins on epidermal MNPs in HIV uptake.

This report on understudied subsets of macrophages and DCs in tissues and their roles in HIV transmission is interesting and offers insights that are potentially important to our general understanding of HIV transmission. However, there are several points that need to be addressed in order to reveal the relevance of these findings. Some specific questions and concerns are the following.

Major concerns:

1. Whether these MNPs are productively infected is not definitive. Since myeloid cells are well known to capture and retain infectious HIV for a while, HIV particles in the culture supernatants maybe those which are slowly released from viral containing compartments instead of newly produced particles. A proper control such as cells with HIV inhibitors (RT, IN etc) with a different method of detection (ELISA or p24 intracellular staining) need to be tested to show productive infection. The same holds true for 96hr trans-infection assays.
2. From the data presented here, it is difficult to have a sense of abundance and localization of MNPs in tissues. The percentage of CD45+ HLA-DR+ among total cells is not shown in Fig 1B. In addition, the immunofluorescence images are not clear enough to see tissue structure and localization of MNPs. It would be helpful to include immunohistochemistry images showing localization of MNP subsets (in particular, langerin+ cDC2 and CD1c+ MDDCs) in tissues possibly with a quantitative measure such as X cells/mm² of a tissue section. The amount of HIV particles in whole semen is not high (median viral load 1.1×10^4 copies/ml, PMID: 9223532). And the efficiency of HIV capture by MNPs and trans-infection from HIV-laden MNPs to CD4 T cell line is not high even at moi 1. It would not be surprising to see even lower trans-infection if primary T cells are used. Therefore, one would wonder how it would be possible for such rare MNPs to contribute to HIV transmission in vivo. Additional data would make the authors' conclusions convincing. At least this point should be discussed.
3. Expression of molecules relevant for DC function and DC-HIV interaction is analyzed by mRNA and suffice protein (Fig. 2C, D). Although mRNA and protein expression levels match in most of the proteins, there are several exceptions. For example, CD80, CD83 and CXCR4, where mRNA is abundant but proteins are poorly detected on the surface. The authors should discuss this discrepancy. Is it because these proteins never get to the surface? Alternatively, staining of these proteins is not efficient due to affinity of antibodies or dim fluorophores? If the latter, is there a better way to represent protein expression data?
4. It is interesting that CD1c+ DCs are more susceptible to HIV than CD1c- MDMs, since DCs have been thought to be less permissive to HIV infection than macrophages. APOBEC3G (Fig 2B) should not account for this difference since Vif+ HIV is used. Have the authors tested other restriction factors such as antiviral de-phosphorylated SAMHD1 expression in MNPs?
5. Fig 4b is not clear to see which cells are HIV RNA+. Clear images would be needed. In the upper panels, RNA+ signals are like huge aggregates which are as large as a cell. Is this a technical issue or many HIV particles are accumulated in virus containing compartments?
6. In Fig 4c, there seems to be decent reduction in MFI of p24 staining, suggesting the number of HIV-1 particles captured per cell may be profoundly reduced by anti-Siglec1 antibodies, although % of p24+ cells is not dramatically reduced in CD1c+ DCs. It would be worth showing geo MFI as well, and more sensitive and quantitative method such as ELISA should be considered. In the same experiment (and Fig 4c), MNPs from different tissues show considerable differences. Is the variability due to donor-to-donor variation or tissue-to-tissue variation even though these MNPs express the same set of surface markers?
7. The authors have used a t-test or the Wilcoxon test to compare two groups where there are more

than two groups. ANOVA or Kruskal-Wallis test with a post-hoc test ought to be used instead. Plus, detailed methods should be described in Method (software, normality test and so on).

Other concerns and comments:

1. MDM and MDDC are widely used to represent MDM/MDDC generated in vitro. Since in vitro MDM/MDDC were also used in this study, it is important to clearly distinguish in vivo (ex vivo) CD1c+ MDM/MDDC from in vitro MDM/MDDC in the main text to avoid confusion.

2. It is interesting that CCR7+ DCs emigrate from tissues spontaneously. In case of HIV mucosal transmission, it is postulated that HIV or other stimuli, such as bacterial components leaked through damaged epithelium due to local trauma, would lead to activation of DCs and migration of HIV-laden DCs. Have the authors tested to stimulate tissues with TLR agonists (LPS etc.) or HIV to profile and quantify migrating cells?

3. Does sqrt (Fig 4f, 5d and 6c) stand for square root? Is there any reason why raw % numbers were converted in this unusual way?

4. HIV particles in semen have been postulated to be opsonized, and the role of opsonization in HIV transmission has been debated. It would be nice if the authors would provide with some comments on opsonization including expression of complement receptors on MNPs.

5. It has been reported that langerin on Langerhans cells restricts HIV cis-infection, but, in this study, it is implied that langerin on cDC2 promotes HIV cis-infection of cDC2 (Fig 6f). Although this hypothesis needs to be tested in further experiments using anti-langerin antibodies, discussion about the differential role of langerin on LCs and cDC2 would be helpful.

6. Line 535, CD1a and CD141 data cannot be found anywhere.

7. Lines 542-544, discussion on APOBEC3G is unclear. A3G has not been thought to affect particle production, and, to begin with, Vif+ HIV is used in this study.

Reviewer #3:

Remarks to the Author:

The manuscript by Rhodes et al. provides a precise and detailed characterization of how the mononuclear phagocytes from the colon and rectum, including macrophages, dendritic cells (DCs) and langerhans cells (LCs), interact with HIV. They first describe how these cells differ from the ones of the skin abdomen using flow cytometry and transcriptomics, redefining the similarities of previously described subsets to either macrophages or DCs. They then analyze the expression of HIV receptors in these various cell types as well as their infectivity by HIV ex vivo and in situ. They observe that although both macrophages and DCs express Siglec-1 and can thus internalize the virus, this one mainly infects DCs. Finally they identify mucosal monocyte-derived DCs and Langerin-expressing cDC2s as the cells that most efficiently transmit HIV to T lymphocytes.

All experiments are very well designed and interpreted and the article clearly written. Although people that are not familiar with the complex nomenclature of the markers that define mononuclear phagocytes may have difficulties to follow (maybe the authors should make some effort to simplify this if possible), I feel this manuscript will be of great interest for immunologists, virologists and more generally, for all clinical researchers working on HIV.

I would like to suggest few clarifications prior to publication:

1- Why do the authors constantly talk about sub-epithelial macrophages and DCs? Indeed, in mucosal

tissue, only some subsets of macrophages and DCs are just below the epithelium, whereas others are located further down, closer to crypts. As the authors digest the tissue, it is unlikely that they can distinguish between these various cell types. This point should thus be clarified: the authors do not know the sub-tissular distribution of the various phagocytes they study.

2- The authors observe that SIGLEC1 macrophages take up important amounts of virus, why don't they discuss the possibility that these cells can be virus reservoirs that transmit viral particles to DCs for them to migrate & transmit them to T cells in lymph nodes?

3- Related to points 1 & 2, it would be great, if possible, to try to infect directly the tissue sections and then analyze them by both tissue staining and flow cytometry. This would allow introducing the temporal variable into infection experiments, which should provide a lot of information on putative mechanisms. Indeed, as an example, maybe DCs have more virus in situ as compared to other cells because they do not secrete them rather than because they internalize them more. Doing experiments at steady state prevents proper interpretation of these data. This should at least be clarified.

Reviewer Rebuttal:

Reviewer #1:

Reviewer: 1) While the comprehensive description of sub-epithelial MNP subsets with regard to cell surface receptors and the ability to deliver virus to T cells is appreciated, the study does not represent a fundamentally new finding/concept. Rather, this is an extension of the previous work on epidermal MNPs in anogenital and colorectal tissues from this laboratory (reported in Nature Communications last year) to other parts of the same tissues.

Response: We acknowledge that this manuscript is an extension of our previous Nature Communications manuscript, however we believe the study is equally important. Together, these two manuscripts map out the full array of HIV target mononuclear phagocytes (MNP) present in the epithelium and sub-epithelium* of all human anogenital and colorectal tissues that are the actual sites of initial HIV acquisition during sexual transmission. Furthermore, we have repeated all key observations using a clinically relevant transmitted/founder strain. The focus of the sub-epithelium in this study is important given that HIV transmission is strongly correlated with trauma and inflammation meaning the virus may well come into direct contact with these cell populations. We have now identified the three specific key MNPs that probably play the most significant role in transmitting HIV to CD4 T cells. This represents a significant advance in our understanding of HIV transmission. It is translationally significant as this information may aid in the development of (i) better blocking strategies to enhance PrEP regimens and (ii) vaccine design, as these are the cells that will likely need to be targeted to deliver HIV antigen to CD4 tells to elicit an effective immune response.

*(NB, We use the word sub-epithelial to refer all tissue layers below the mucosal epithelial surface. Most tissues are not colorectal but skin and anogenital. For skin (abdomen, labia, foreskin, glans penis, anal verge) it refers to the dermis, in Type II mucosal tissue (vagina, ectocervix, fossa naviculars, anal canal) in refers to lamina propria and in Type I mucosal tissue (colon, rectum, endocervix and penile urethra) it refers so lamina propria, muscularis mucosa and submucosa -see response to Reviewer 3).

Reviewer: 2) Fig 4B is not convincing. In top right panel, all HIV signals are associated with CD14-negative but not CD14-positive cells. In bottom right CD14+CD1c+ cells are shown, but the level of HIV associated with these cells are very low compared to those observed in top right, contradicting the outcomes of Fig 4A, which suggest that CD14+CD1c- cells are most efficient in virus uptake.

Response: Flow cytometry is a highly quantitative technique whereas the representative microscopy images in Figure 4B is qualitative only. Figure 4A represents thousands of individual cells derived from 7 individual donors and the results are statistically significant. Figure 4B was only intended to show representative images of CD14 expressing cell subsets to confirm that cells can interacting with HIV *in situ*. We acknowledge that Figure 4B contained a poor choice of image for the reasons stated by the reviewer and we have now carried out further staining using human penile urethra to obtain more appropriate representative images (see new Figure 4B).

Reviewer: 3) In the Fig 4F and 5D experiments, the authors co-cultured the GFP reporter T cell line (JLTR) with infected MNPs for 96 h and used the GFP-positive cell number as the readout for the ability of MNPs to transfer HIV to CD4+ T cells. There are three concerns with this set of experiments.

a) The ability of virus transfer to T cells is affected not only by the ability of MNPs to capture/propagate HIV-1 but also by T cell adhesion molecules that bind MNP surface molecules. Since T cell lines may not fully recapitulate surface protein profiles of primary T cells, to test the true ability of MNPs to deliver HIV-1 to “T cells”, the co-culture experiments should be repeated using primary T cells.

Response: Using primary T cells for this assay is very difficult because they must be stained for p24 expression at 96 hours after infection. We have previously successfully done this in our previous Nature Communications paper (Bertram et al., 2019) for a single subset of primary CD11c⁺ epidermal DCs (using clinical transmitted/founder isolates) and also in our Journal of Immunology (Nasr et al., 2014) manuscript using primary Langerhans cells (PMID: 25070850). However, to carry out these assays using every sub-epithelial MNP subsets (6 in total) across at least 4-6 donors to achieve statistical significance could literally take years and would likely prove impossible for some MNP subsets given the very small cell numbers we can isolate (e.g. langerin⁺ cDC2 and cDC1). Given that we have already shown that primary *ex vivo* epidermal CD11c⁺ dendritic cells (epidermal cDC2) and Langerhans cells both transfer HIV to primary CDT cells in a similar fashion to our JLTR assay, we have now taken the total CD14 expressing cell population derived from abdominal tissue and conducted comparisons using JLTRs and primary T cells concurrently in three donors (**new Figure 4G**). We have used bulk population to generate a high enough cell yield to reliably carry out the assay. In all donors the virus was transferred to both cell types although higher levels of HIV infectivity were detected in JLTRs presumably due to their very high levels of CCR5 expression. We presume most of the transfer from these CD14 MNPs is from monocyte-derived dendritic cells (MDDC) because transfer was greater from MDDCs than monocyte-derived macrophages (MDM). Thus, our group has now shown similar results using both assays for multiple *ex vivo* derived MNP populations in three individual manuscripts.

b) Under the condition used in this study, virus spread from T cells to T cells can occur. The efficiency of T cell-to-T cell spread can be differentially altered in the presence of different MNPs (directly or indirectly, e.g., via cytokine release). Therefore, the difference seen between MNP subsets in this assay may not necessarily represent the difference in efficiency of MNP-to-T cell HIV transfer.

Response: The reviewer is concerned that MNP cytokine release may be enhancing T cell to T cell transfer rather than, or as well as, MNP transfer. We believe this is unlikely to be substantial especially at the early phase (2 hours) where transfer is from DC caves (i.e. extracellular and less likely to be a powerful inducer of cytokines). However, if this mechanism does contribute, it clearly varies amongst MNP subsets in targeting spread to and between CD4 T cells and therefore it is no less relevant to transmission. We have now added this as a discussion point in the discussion:

“It is of note that under the conditions used in this study, virus spread from T cells to T cells may occur. It is possible that the efficiency of T cell-to-T cell spread may be differentially altered in the presence of different MNPs (e.g. via cytokine release). We believe this is unlikely to be substantial especially in the context of early phase (2 hour) transfer which is from MNP intracellular caves which is unlikely to be a powerful inducer of cytokine release. However, if this mechanism does contribute, it clearly varies amongst MNP subsets in targeting spread to CD4 T cells and therefore it is no less relevant to transmission.”

Reviewer #2:

Reviewer: 1) Whether these MNPs are productively infected is not definitive. Since myeloid cells are well known to capture and retain infectious HIV for a while, HIV particles in the culture supernatants maybe those which are slowly released from viral containing compartments instead of newly produced particles. A proper control such as cells with HIV inhibitors (RT, IN etc) with a different method of detection (ELISA or p24 intracellular staining) need to be tested to show productive infection. The same holds true for 96hr trans-infection assays.

Response: As we mentioned in the Discussion section of the manuscript, we found that it was not possible to measure direct infection of individual subsets by flow cytometry as a substantial majority of these cells die in culture after tissue isolation. However, we have now isolated a bulk population of CD14+ cells and cDC2 (lang+ and -) and cultured these with and without AZT using three independent donors (new **Figure 5D and 6G**). As expected, AZT treated cells produced little or no infectious virions measured using our TZMBL1 assay. Note that all AZT was washed off after 48 hours of infection and did not directly inhibit HIV infection in as per the appropriate control (see below) and in our previous papers. We have already shown in multiple manuscripts (PMID: 31227717, 25070850, 14630806) that transfer of HIV to CD4+ T cells rapidly declines with time and that within 24 hours no transfer occurs. Later (72 – 96 hours) a second phase of transfer occurs via *de novo* infection as newly formed virions bud off from the surface. This shows that using our system, later stage transfer does not occur as a result of virions held on the cell surface or contained within intracellular compartments.

Reviewer: 2) From the data presented here, it is difficult to have a sense of abundance and localization of MNPs in tissues. The percentage of CD45+ HLA-DR+ among total cells is not shown in Fig 1B. In addition, the immunofluorescence images are not clear enough to see tissue structure and localization of MNPs. It would be helpful to include immunohistochemistry images showing localization of MNP subsets (in particular, langerin+ cDC2 and CD1c+ MDDCs) in tissues possibly with a quantitative measure such as X cells/mm² of a tissue section. The amount of HIV particles in whole semen is not high (median viral load 1.1x10⁴ copies/ml, PMID: 9223532). And the efficiency of HIV capture by MNPs and trans-infection from HIV-laden MNPs to CD4 T cell line is not high even at moi 1. It would not be surprising to see even lower trans-infection if primary T cells are used. Therefore, one would wonder how it would be possible for such rare MNPs to contribute to HIV transmission in vivo. Additional data would make the authors' conclusions convincing. At least this point should be discussed.

Response: Using the current extensive data set shown in Figure 1B (which took 5 years to acquire) it is not possible to determine the percentage of CD45+ HLA-DR+ among total cells because we used CD45 magnetic bead enrichment prior to flow cytometry analysis. However, at the reviewers request we have now quantitatively measured MNP subsets as cells/mm² of a tissue section (new **Figure 1C**)

and defined their average distance from basement membrane using 5 inner foreskin donors. The relative proportions of MNP subsets determined by microscopy is consistent with the flow cytometry data (new **Figure 1C**). We found little difference in the distance from basement membrane between MNP subsets (including CD14⁺ MDDCs) so have chosen not to include this data in the new manuscript but show it below. Should the reviewer/editor want this included in the manuscript we are happy to do so.

We have also now added a section in the discussion as to how it might be possible for such rare MNPs to contribute to HIV transmission *in vivo*. Regarding the question “*one would wonder how it would be possible for such rare MNPs to contribute to HIV transmission in vivo?*” we believe that by transferring virus to CD4 T cells MNPs amplify infection. In our previous Nature Communications paper (Bertram et al., 2019) we showed that in a healthy mucosa CD11c⁺ epidermal cDC2 are likely to be the first cells that HIV encounters during transmission and that these cells transmit HIV to CD4 T cells in a preferential fashion over Langerhans cells. However, most transmission probably occurs in the context of mucosal inflammation and also microtrauma, meaning that the first cells HIV comes into contact with, especially in the latter circumstance, are likely to be MNP in the underlying lamina propria/dermis. Thus, the mapping out of the relative proportions on MNPs in the tissues where transmission occurs is important and even more so their relative ability to transmit HIV to CD4 T cells. In the case of langerin⁺ cDC2, although these cells are rare, they are by far the most efficient at transfer to CD4 T cells. This high capacity for viral transfer is what we hypothesise makes it possible for them to contribute to transmission. As requested, we have now added a sentence to the discussion of the modified manuscript:

“Importantly, we found that langerin-expressing cDC2 were much more efficient at HIV uptake after 2 hours than their non-langerin-expressing counterparts and were correspondingly much more efficient at transferring the virus to CD4 T cells at the same early time point. In fact, these cells were the most efficient of all MNPs at transfer of HIV to CD4 T cells, meaning that despite their relatively lower frequency compared to other MNP cell types, they are nevertheless likely to be key players in HIV transmission.”

Reviewer: 3) Expression of molecules relevant for DC function and DC-HIV interaction is analyzed by mRNA and surface protein (Fig. 2C, D). Although mRNA and protein expression levels match in most of the proteins, there are several exceptions. For example, CD80, CD83 and CXCR4, where mRNA is abundant but proteins are poorly detected on the surface. The authors should discuss this discrepancy. Is it because these proteins never get to the surface? Alternatively, staining of these proteins is not efficient due to affinity of antibodies or dim fluorophores? If the latter, is there a better way to represent protein expression data?

Response: We have previously published our extensive optimisation of enzymatic cleavage of surface markers and our ability to detect the surface expression of all key DC markers in this manuscript by flow cytometry including CD80, CD83 and CXCR4 (PMID: 28270408). In the same paper we show that cells begin to undergo a process of maturation as a result of tissue extraction. CD80, CD83 and CXCR4 are all upregulated in the maturation process. Therefore, the most likely explanation for the difference in mRNA and surface protein levels is that at the time point we measured surface expression (immediately after isolation) changes in gene expression had not yet translated to surface expression. We have now added a sentence mentioning this in the results section of the modified manuscript.

“Discrepancies between mRNA and surface expression levels were detected in molecules upregulated due to MNP maturation (CD80, CD83 and CXCR4). This is most likely due to the fact that the process of tissue isolation triggers maturation as shown previously⁴⁰. Thus, at the time point we measured surface expression (immediately after isolation) changes in gene expression had not yet translated to surface expression.”

Reviewer: 4) *It is interesting that CD1c+ DCs are more susceptible to HIV than CD1c- MDMs, since DCs have been thought to be less permissive to HIV infection than macrophages. APOBEC3G (Fig 2B) should not account for this difference since Vif+ HIV is used. Have the authors tested other restriction factors such as antiviral de-phosphorylated SAMHD1 expression in MNPs?*

Response: We have now measured intracellular dephosphorylated SAMHD1 levels in all human tissue MNP subsets derived from abdominal dermis using intracellular flow cytometry in four individual donors. As per the literature, we saw a trend of SAMHD1 being expressed more highly in DCs than macrophages and significantly higher in cDC1 (which also express very low levels of CCR5 and are not permissive to HIV infection). SAMHD1 was expressed slightly more highly in *ex vivo* CD14⁺CD1c⁺ MDDCs than in *ex vivo* CD14⁺CD1c⁻ MDMs although this was not statistically significant. The data is shown below in **new Figure 2B**. We have now modified our statement regarding this in the discussion to *“They also expressed lower levels of the HIV restriction factor APOBEC3G which could further account for the higher levels of secreted virions by these cells although the HIV Vif protein is known to inhibit the function of this protein. Expression of dephosphorylated SAMHD1 by flow cytometry was not significantly different between the two cell types”*

Reviewer: 5) *Fig 4b is not clear to see which cells are HIV RNA+. Clear images would be needed. In the upper panels, RNA+ signals are like huge aggregates which are as large as a cell. Is this a technical issue or many HIV particles are accumulated in virus containing compartments?*

Response: We have carried out further staining using human penile urethra to obtain more convincing images.

Reviewer: 6) *In Fig 4c, there seems to be decent reduction in MFI of p24 staining, suggesting the number of HIV-1 particles captured per cell may be profoundly reduced by anti-Siglec1 antibodies, although % of p24+ cells is not dramatically reduced in CD1c+ DCs. It would be worth showing geo MFI as well, and more sensitive and quantitative method such as ELISA should be considered.*

Response: We are confused by this comment. Figure 4C shows the MFI of Siglec-1 expression not p24. The percentage of blocking of HIV-1 uptake by Siglec-1 is shown in Figure 4E which shows that the

percentage of p24 positive cells (not p24 MFI) was reduced. Could we respectfully ask the reviewer to more clearly state their concern?

In the same experiment (and Fig 4c), MNPs from different tissues show considerable differences. Is the variability due to donor-to-donor variation or tissue-to-tissue variation even though these MNPs express the same set of surface markers?

Response: We are unsure if the differences shown are donor or tissue specific. In order to answer this question, we would have to repeat the experiment 2-3 more times for each tissue which would likely take 18 months or longer. We would rather leave this question unanswered.

Reviewer: 7). The authors have used a t-test or the Wilcoxon test to compare two groups where there are more than two groups. ANOVA or Kruskal-Wallis test with a post-hoc test ought to be used instead. Plus, detailed methods should be described in Method (software, normality test and so on).

Response: We have now carried ANOVA as requested and have included detailed methods in the methods section. This has not affected the statistical significance of the results.

Other concerns and comments:

Reviewer: 1) MDM and MDDC are widely used to represent MDM/MDDC generated in vitro. Since in vitro MDM/MDDC were also used in this study, it is important to clearly distinguish in vivo (ex vivo) CD1c+ MDM/MDDC from in vitro MDM/MDDC in the main text to avoid confusion.

Response: We have modified the manuscript to ensure every time the acronyms MDM or MMDC are referred to it is clear whether we are referring to *ex vivo* or *in vitro* derived.

Reviewer: 2) It is interesting that CCR7+ DCs emigrates from tissues spontaneously. In case of HIV mucosal transmission, it is postulated that HIV or other stimuli, such as bacterial components leaked through damaged epithelium due to local trauma, would lead to activation of DCs and migration of HIV-laden DCs. Have the authors tested to stimulate tissues with TLR agonists (LPS etc.) or HIV to profile and quantify migrating cells?

Response: At the reviewer's suggestion we have carried out this experiment and added it to the results as Figure 3E. "Finally, as it is postulated that HIV or other stimuli, such as bacterial components leaked through damaged epithelium due to local trauma, would lead to activation of DCs and migration of HIV-laden DCs, we stimulated abdominal skin tissue with the TLR agonist imiquimod and showed that this significantly increased the number of *ex vivo* MDDCs that migrated out of the tissue (**Figure 3E**)."

Reviewer: 3) Does sqrt (Fig 4f, 5d and 6c) stand for square root? Is there any reason why raw % numbers were converted in this unusual way?

Response: Yes, sqrt stands for square root. We converted the data this way for greater ease of visualisation however it does not change the statistical significance of the results.

Reviewer: 4) HIV particles in semen have been postulated to be opsonized, and the role of opsonization in HIV transmission has been debated. It would be nice if the authors would provide with some comments on opsonization including expression of complement receptors on MNPs.

Response: We have analysed our RNAseq data and have found no differences in the expression of complement receptor expression between any MNP subsets.

Reviewer: 5) It has been reported that langerin on Langerhans cells restricts HIV cis-infection, but, in this study, it is implied that langerin on cDC2 promotes HIV cis-infection of cDC2 (Fig 6f). Although this hypothesis needs to be tested in further experiments using anti-langerin antibodies, discussion about the differential role of langerin on LCs and cDC2 would be helpful.

Response: The Geijtenbeek group have reported that langerin on Langerhans cells restricts HIV cis infection (PMID: 17334373). However, to our knowledge they are the only group to have shown this and we and others do not find the same results. Indeed, we published the exact opposite result by showing that blocking HIV's ability to bind langerin on Langerhans cells (using both soluble langerin and a langerin mAb) directly blocks their ability to transmit HIV to CD4 T cells (PMID: PMID: 25070850). It is of note that Geijtenbeek group used trypsin to isolate the cells in their study which we have shown cleaves multiple HIV binding receptors including CD4 and multiple CLRs (PMID: 28270408). Experiments such as these should therefore use collagenase Type IV to isolate the cells from tissue.

Reviewer: 6) Line 535, CD1a and CD141 data cannot be found anywhere.

Response: We apologise for this oversight. This data has now been included in Figure 2C.

Reviewer: 7) Lines 542-544, discussion on APOBEC3G is unclear. A3G has not been thought to affect particle production, and, to begin with, Vif+ HIV is used in this study.

Response: This has now been mentioned in the discussion.

Reviewer #3:

We thank the reviewer for their supportive words.

Reviewer: 1) Why do the authors constantly talk about sub-epithelial macrophages and DCs? Indeed, in mucosal tissue, only some subsets of macrophages and DCs are just below the epithelium, whereas others are located further down, closer to crypts. As the authors digest the tissue, it is unlikely that they can distinguish between these various cell types. This point should thus be clarified: the authors do not know the sub-tissular distribution of the various phagocytes they study.

Response: We use the word sub-epithelial to refer all tissue layers below the mucosal epithelial surface. Most tissues are not colorectal but skin and anogenital. For skin (abdomen, labia, foreskin, glans penis, anal verge) it refers to the dermis, in Type II mucosal tissue (vagina, ectocervix, fossa navicularis, anal canal) it refers to lamina propria and in Type I mucosal tissue (colon, rectum, endocervix and penile urethra) it refers so lamina propria, muscularis mucosa and submucosa.

Reviewer: 2) The authors observe that SIGLEC1 macrophages take up important amounts of virus, why don't they discuss the possibility that these cells can be virus reservoirs that transmit viral particles to DCs for them to migrate & transmit them to T cells in lymph nodes?

Response: We have added this point to the discussion as requested:

"It is of note that ex vivo CD14+ tissue resident macrophages and MDMs were able to take up HIV (albeit significantly less so than MDDCs). The possibility therefore exists that these cells act as virus reservoirs that transmit viral particles to DCs which could then migrate and transmit the virus to T cells."

Reviewer: 3) Related to points 1 & 2, it would be great, if possible, to try to infect directly the tissue sections and then analyze them by both tissue staining and flow cytometry. This would allow introducing the temporal variable into infection experiments, which should provide a lot of information on putative mechanisms. Indeed, as an example, maybe DCs have more virus in situ as compared to other cells because they do not secrete them rather than because they internalize them more. Doing experiments at steady state prevents proper interpretation of these data. This should at least be clarified.

Response: This is a nice idea but not logistically possible. In order to isolate enough cells to carry out our *ex vivo* experiments we use abdominal skin explants that are the size of 2 - 4 sheets of A4 paper (see photo). To grow a virus stock large enough to directly infect explants of this size is impossible. Even if it was possible, it would be impossible to detect enough directly infected cells by flow cytometry as cells require 72-96 hours of infection to be able to directly detect intracellular p24.

Figures 4B and 6B show cells in infected tissue biopsies containing HIV virus particles within 2 hours of topic infection detected using our RNA scope protocols.

Reviewers' Comments:

Reviewer #2:

Remarks to the Author:

The authors were quite responsive to prior critiques with more data and addressed the concerns raised by the reviewers. There is no further concerns. Here are some minor points to clarify.

It is much appreciated that the authors have included the data on expression of SAMHD1 in MNPs though, the key is whether SAMHD1 is phosphorylated (no restriction) or not (antiviral). The antibody used (clone I19-18 from Merck/Sigma-Aldrich) does not seem to distinguish the difference. If this is the case, the authors should describe this limitation.

As for the point #6: Sorry about the typing error. The question was about Fig. 4E. The authors show the effect of anti-Siglec1 antibody on HIV capture by calculating reduction in the percentage of p24+ cells and find that the blocking efficiency of HIV capture in ex vivo DCs was only modest, while ex vivo DCs showed similar capacity to other MNPs in transferring HIV to target cells (Fig 4F). The authors therefore conclude that "Siglec-1 expression does not explain why ex vivo MDDCs cells take up HIV more efficiently than other CD14-expressing cells". In Fig 4E, however, the MFI of p24 signals looks dramatically reduced within p24+ population also in ex vivo DCs upon treatment with anti-Siglec1 antibody. The question is: if the authors calculate the effect of Siglec1 blocking on HIV capture based on changes in the MFI of p24 staining (i.e. HIV particle number per cell), would that change Fig 4F and their conclusion on the role of Siglec1?

Reviewer #3:

Remarks to the Author:

The manuscript is considerably improved and acceptable for publication.

NCOMMS-20-19712 Reviewer Rebuttal:

Reviewer #1:

No comments provided

Reviewer #2:

Reviewer: The authors were quite responsive to prior critiques with more data and addressed the concerns raised by the reviewers. There is no further concerns. Here are some minor points to clarify.

Response: We are delighted that the reviewer feels there is no further concerns and thank them for the constructive critical feedback. Please find our clarification of the two minor points below:

Reviewer: 1) It is much appreciated that the authors have included the data on expression of SAMHD1 in MNPs though, the key is whether SAMHD1 is phosphorylated (no restriction) or not (antiviral). The antibody used (clone I19-18 from Merck/Sigma-Aldrich) does not seem to distinguish the difference. If this is the case, the authors should describe this limitation.

Response: We have now added the following sentence to the discussion: "It is of note that the SAMHD1 antibody clone we used did not distinguish whether the SAMHD1 was phosphorylated (no restriction) or not (antiviral)."

Reviewer: 2) As for the point #6: Sorry about the typing error. The question was about Fig. 4E. The authors show the effect of anti-Siglec1 antibody on HIV capture by calculating reduction in the percentage of p24+ cells and find that the blocking efficiency of HIV capture in ex vivo DCs was only modest, while ex vivo DCs showed similar capacity to other MNPs in transferring HIV to target cells (Fig 4F). The authors therefore conclude that "Siglec-1 expression does not explain why ex vivo MDDCs cells take up HIV more efficiently than other CD14-expressing cells". In Fig 4E, however, the MFI of p24 signals looks dramatically reduced within p24+ population also in ex vivo DCs upon treatment with anti-Siglec1 antibody. The question is: if the authors calculate the effect of Siglec1 blocking on HIV capture based on changes in the MFI of p24 staining (i.e. HIV particle number per cell), would that change Fig 4F and their conclusion on the role of Siglec1?

Response: We have analysed the MFI of p24 staining on all three subsets of CD14 expressing cells and have found no difference in the results and we therefore have not changed our conclusion regarding the role of Siglec1 in HIV uptake in ex vivo MDDCs.

Reviewer #3:

Reviewer: The manuscript is considerably improved and acceptable for publication.

Response: We are delighted that the reviewer feels the manuscript is now acceptable for publication and thank them for the constructive critical feedback.